# Boomerang: Local sampling on image manifolds using diffusion models

**Lorenzo Luzi** *enzo@rice.edu*
*Rice University*

**Paul M. Mayer** *pmm3@rice.edu*
*Rice University*

**Josue Casco-Rodriguez** *jc135@rice.edu*
*Rice University*

**Ali Siahkoohi** *alisk@rice.edu*
*Rice University*

**Richard G. Baraniuk** *richb@rice.edu*
*Rice University*

**Reviewed on OpenReview:** *https://openreview.net/forum?id=NYdThkjNW1*

## Abstract

The inference stage of diffusion models can be seen as running a reverse-time diffusion stochastic differential equation, where samples from a Gaussian latent distribution are transformed into samples from a target distribution that usually reside on a low-dimensional manifold, e.g., an image manifold. The intermediate values between the initial latent space and the image manifold can be interpreted as noisy images, with the amount of noise determined by the forward diffusion process noise schedule. We utilize this interpretation to present Boomerang, an approach for local sampling of image manifolds exploiting the reverse diffusion process dynamics. As implied by its name, Boomerang local sampling involves adding noise to an input image, moving it closer to the latent space, and then mapping it back to the image manifold through a partial reverse diffusion process. Thus, Boomerang generates images on the manifold that are "similar," but nonidentical, to the original input image. We can control the proximity of the generated images to the original by adjusting the amount of noise added. Furthermore, due to the stochastic nature of the partial reverse diffusion process in Boomerang, the generated images display a certain degree of stochasticity, allowing us to obtain ample local samples from the manifold without encountering any duplicates. Boomerang offers the flexibility to work seamlessly with any pretrained diffusion model, such as Stable Diffusion, without necessitating any adjustments to the reverse diffusion process. We present three applications for local sampling using Boomerang. First, we provide a framework for constructing privacy-preserving datasets having controllable degrees of anonymity. Second, we show that using Boomerang for data augmentation increases generalization performance and outperforms state-of-the-art synthetic data augmentation. Lastly, we introduce a perceptual image enhancement framework powered by Boomerang, which enables resolution enhancement.

## 1 Introduction

Generative models have seen a tremendous rise in popularity and applications over the past decade, ranging from image synthesis (Grcić et al., 2021; Dhariwal & Nichol, 2021; Sauer et al., 2022), audio generation (van

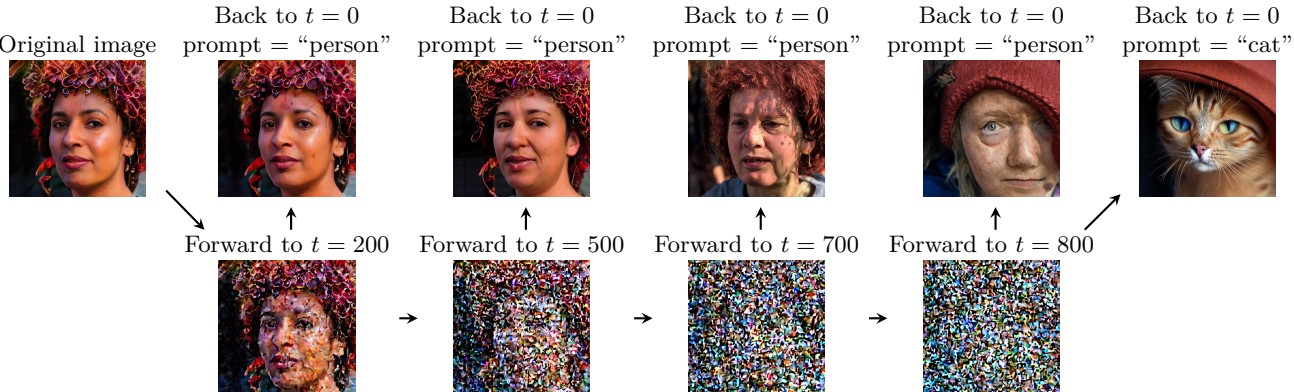

Figure 1: An example using Boomerang via Stable Diffusion (Rombach et al., 2022). Starting from an original image $x_0 \sim p(x_0)$, we add varying levels of noise to the latent variables according to the noise schedule of the forward diffusion process. Boomerang maps the noisy latent variables back to the image manifold by running the reverse diffusion process starting from the reverse step associated with the added noise out of $T = 1000$. The resulting images are local samples from the image manifold, where the closeness is determined by the amount of added noise. Note how, as $t$ approaches to $T$, the content of Boomerang-generated images *strays* further away from the starting image. While Boomerang here is applied to the Stable Diffusion model, it is applicable to other types of diffusion models, e.g., denoising diffusion models (Ho et al., 2020). Additional images are provided in Appendix A.1. A Boomerang Colab demo is available at `https://colab.research.google.com/drive/1PV5Z6b14HYZNx11HCaEVhId-Y4baKXwt`.

den Oord et al., 2016; Klejsa et al., 2019; Kong et al., 2020; 2021), out-of-distribution data detection (Li et al., 2022; Dionelis et al., 2022), and reinforcement learning to drug synthesis (Kingma & Welling, 2014; Goodfellow et al., 2014; Bond-Taylor et al., 2021). One of the key benefits of generative models is that they can generate new samples using training samples from an unknown probability distribution. A family of generative models known as diffusion models have recently gained attention in both the academic and public spotlights with the advent of Dall-E 2 (Ramesh et al., 2022), Imagen (Saharia et al., 2022a), and Stable Diffusion (Rombach et al., 2022).

Generative models estimate the underlying probability distribution, or manifold, of a dataset by learning to sample from this distribution. Sampling with generative models involves mapping samples from a latent distribution, e.g., Gaussian, via the trained generative model to samples from the target distribution, which yields a *global* sampling mechanism. While global sampling is useful for modeling the whole distribution, there are several important problems which require *local* sampling—the ability to produce samples close to a particular data point. The anonymization of data is one application of local sampling in which the identity of subjects in the dataset should be erased while maintaining data fidelity. Another application is data augmentation, which involves applying transformations onto copies of data points to produce new data points (Wong et al., 2016). A third application of local sampling is to remove noise from an image, especially in the cases of severe noise, where traditional denoising methods might fail (Kawar et al., 2021).

Despite their success in global sampling, GANs (Goodfellow et al., 2014), Variational autoencoders (VAEs) (Kingma & Welling, 2014), and Normalizing Flows (NFs) (Rezende & Mohamed, 2015) are not the best candidates for local sampling. GANs do not lend themselves well for local sampling: GAN inversion is required, and finding or training the best methods for GAN inversion is a difficult problem and an ongoing area of research (Karras et al., 2020; Xia et al., 2022; Yu et al., 2020). VAEs and NFs, on the other hand, can both project a data point $x$ into a latent vector $z$ in their respective latent spaces and then re-project $z$ back into the original data space, producing an estimate $x'$ of the original data point. As such, VAEs and NFs can perform local sampling of a data point $x$ by projecting a perturbed version of its corresponding latent vector $z$ back into the original data space. While the straightforward tractability of VAE- or NF-based local sampling is attractive, VAEs and NFs are not state-of-the-art (SOTA) on popular tasks such as image synthesis. For these reasons, we turn to diffusion models, which are SOTA (Dhariwal & Nichol, 2021),

to perform local sampling. For these reasons, we propose local sampling with diffusion models, which are SOTA (Dhariwal & Nichol, 2021) and do not require latent space inversion.

We propose the Boomerang algorithm to enable local sampling of image manifolds using *pretrained* diffusion models. Boomerang earns its name from its principle mechanism—adding noise of a certain variance to push data away from the image manifold, and then using a diffusion model to pull the noised data back onto the manifold. The variance of the noise is the only parameter in the algorithm, and governs how similar the new image is to the original image, as reported by Ho et al. (2020). We apply this technique to three applications: (1) data anonymization for privacy-preserving machine learning; (2) data augmentation; and (3) perceptual enhancement for low resolution images. We show that the proposed local sampling technique is able to: (1a) anonymize entire datasets to varying degrees; (1b) trick facial recognition algorithms; and (1c) anonymize datasets while maintaining better classification accuracy when compared with SOTA synthetic datasets. For data augmentation we: (2a) obtain higher classification accuracy when trained on the Boomerang-augmented dataset versus no augmentation at all; and (2b) outperform SOTA synthetic data augmentation. Finally, we show that Boomerang can be used for perceptual image enhancement. The images generated via local sampling: 3a) have better perceptual quality than those generated with competing methods; 3b) are generated faster than other deep-learning methods trained methods such as the Deep Image Prior (Ulyanov et al., 2020); and 3c) can be used for any desired upsampling factor without needing to train or fine-tune the network.

In Section 2 we discuss the training framework of diffusion models, introducing the forward and reverse processes. In Section 3 we introduce our proposed local sampling method—Boomerang—and provide insights on how the amount of added noise affects the locality of the resulting samples. Finally, we describe three applications (Sections 4 to 6) that Boomerang can be used without any modification to the diffusion model pretraining.

## 2 Diffusion models

Diffusion models sample from a distribution by learning to reverse a forward diffusion process that turns data from the training dataset into realizations of a Gaussian distribution (Ho et al., 2020). The forward diffusion process involves adding Gaussian noise to the input data, $\boldsymbol{x}_0 \sim p(\boldsymbol{x}_0)$, in $T$ steps:

$$\boldsymbol{x}_t := \sqrt{1 - \beta_t} \boldsymbol{x}_{t-1} + \boldsymbol{\epsilon}_t, \quad \boldsymbol{\epsilon}_t \sim \mathcal{N}(\mathbf{0}, \beta_t \boldsymbol{I}), \quad t = 1, \ldots, T, \tag{1}$$

where $\beta_t \in (0, 1)$, $t = 1, \ldots, T$, is the noise variance at step $t$, which is typically chosen beforehand (Song & Ermon, 2020). Since the transition from step $t-1$ to $t$ is defined by a Gaussian distribution in the form of $q(\boldsymbol{x}_t|\boldsymbol{x}_{t-1}) := \mathcal{N}(\sqrt{1 - \beta_t}\boldsymbol{x}_{t-1}, \beta_t \boldsymbol{I})$, the distribution of $\boldsymbol{x}_t$ conditioned on the clean input image $\boldsymbol{x}_0$ can be expressed as a Gaussian distribution,

$$q(\boldsymbol{x}_t|\boldsymbol{x}_0) = \mathcal{N}\left(\sqrt{\alpha_t}\boldsymbol{x}_0, (1 - \alpha_t)\boldsymbol{I}\right), \quad t = 1, \ldots, T, \tag{2}$$

with $\alpha_t = \prod_{i=1}^{t}(1 - \beta_i)$. During training, diffusion models learn to reverse the forward process by starting at $t = T$ with a sample from the standard Gaussian distribution $\boldsymbol{x}_T \sim \mathcal{N}(\mathbf{0}, \boldsymbol{I})$. The reverse process is defined via a Markov chain over $\boldsymbol{x}_0, \boldsymbol{x}_1, \ldots, \boldsymbol{x}_T$ such that

$$\boldsymbol{x}_{t-1} := f_\phi(\boldsymbol{x}_t, t) + \boldsymbol{\eta}_t, \quad \boldsymbol{\eta}_t \sim \mathcal{N}(\mathbf{0}, \bar{\beta}_t \boldsymbol{I}), \quad t = 1, \ldots, T. \tag{3}$$

In the above expression, $f_\phi(\boldsymbol{x}_t, t)$ is parameterized by a neural network with weights $\boldsymbol{\phi}$, and $\bar{\beta}_t \boldsymbol{I}$ denotes the covariances at step $t$. Equation (3) represents a chain with transition probabilities defined with Gaussian distributions with density

$$p_\phi(\boldsymbol{x}_{t-1}|\boldsymbol{x}_t) := \mathcal{N}(f_\phi(\boldsymbol{x}_t, t), \bar{\beta}_t \boldsymbol{I}), \quad t = 1, \ldots, T. \tag{4}$$

The covariance $\bar{\beta}_t \boldsymbol{I}$ in different steps can be also parameterized by neural networks, however, we follow Luhman & Luhman (2022) and choose $\bar{\beta}_t = \frac{1 - \alpha_{t-1}}{1 - \alpha_t} \beta_t$, that matches the forward posterior distribution when conditioned on the input image $q(\boldsymbol{x}_{t-1}|\boldsymbol{x}_t, \boldsymbol{x}_0)$ (Ho et al., 2020).

To ensure the Markov chain in Equation (3) reverses the forward process (Equation (1)), the parameters $\boldsymbol{\phi}$ are updated such that the resulting image at step $t = 0$ via the reverse process represents a sample from the target distribution $p(\boldsymbol{x}_0)$. This can be enforced by maximizing—with respect to $\boldsymbol{\phi}$—the likelihood $p_\phi(\boldsymbol{x}_0)$ of training samples where $\boldsymbol{x}_0$ represents the outcome of the reverse process at step $t = 0$. Unfortunately, the density $p_\phi(\boldsymbol{x}_0)$ does not permit a closed-form expression. Instead, given the Gaussian transition probabilities defined in Equation (4), the joint distribution over all the $T + 1$ states can be factorized as,

$$p_\phi(\boldsymbol{x}_0, \dots, \boldsymbol{x}_T) = p(\boldsymbol{x}_T) \prod_{t=1}^T p_\phi(\boldsymbol{x}_{t-1}|\boldsymbol{x}_t), \quad p_T(\boldsymbol{x}_T) = \mathcal{N}(\boldsymbol{0}, \boldsymbol{I}), \tag{5}$$

with all the terms on the right-hand-side of the equality having closed-form expressions. To obtain a tractable expression for training diffusion models, we treat $\boldsymbol{x}_1, \dots, \boldsymbol{x}_T$ as latent variables and use the negative evidence lower bound (ELBO) expression for $p_\phi(\boldsymbol{x}_0)$ as the loss function,

$$\mathcal{L}(\boldsymbol{\phi}) := \mathbb{E}_{p(\boldsymbol{x}_0)} \mathbb{E}_{q(\boldsymbol{x}_1, \dots, \boldsymbol{x}_T | \boldsymbol{x}_0)} \left[ -\log p_T(\boldsymbol{x}_T) - \sum_{t=1}^T \log \frac{p_\phi(\boldsymbol{x}_{t-1}|\boldsymbol{x}_t)}{q(\boldsymbol{x}_t|\boldsymbol{x}_{t-1})} \right] \tag{6}$$
$$\geq \mathbb{E}_{p(\boldsymbol{x}_0)} \left[ -\log p_\phi(\boldsymbol{x}_0) \right].$$

After training, new global samples from $p_\phi(\boldsymbol{x}_0) \approx p(\boldsymbol{x}_0)$ are generated by running the reverse process in Equation (3) starting from $\boldsymbol{x}_T \sim \mathcal{N}(\boldsymbol{0}, \boldsymbol{I})$. Due to the stochastic nature of the reverse process, particularly, the additive noise during each step, starting from two close initial noise vectors at step $T$ does not necessarily lead to close-by images on the image manifold. The next section describes our proposed method for local sampling on the image manifold.

## 3 Boomerang method

Our method, Boomerang, allows one to locally sample a point $\boldsymbol{x}_0'$ on an image manifold $\mathcal{X}$ close to a point $\boldsymbol{x}_0 \in \mathcal{X}$ using a pretrained diffusion model $f_\phi$. Since we are mainly interested in images, we suppose that $\boldsymbol{x}_0$ and $\boldsymbol{x}_0'$ are images on the image manifold $\mathcal{X}$. We control how close to $\boldsymbol{x}_0$ we want $\boldsymbol{x}_0'$ to be by setting the hyperparameter $t_{\text{Boom}}$. We perform the forward process of the diffusion model $t_{\text{Boom}}$ times, from $t = 0$ to $t = t_{\text{Boom}}$ in Equation (1), and use $f_\phi$ to perform the reverse process from $t = t_{\text{Boom}}$ back to $t = 0$. If $t_{\text{Boom}} = T$ we perform the full forward diffusion and hence lose all information about $\boldsymbol{x}_0$; this is simply equivalent to globally sampling from the diffusion model. We denote this partial forward and reverse process as $B(\boldsymbol{x}_0, t_{\text{Boom}}) = \boldsymbol{x}_0'$ and call it *Boomerang* because $\boldsymbol{x}_0$ and $\boldsymbol{x}_0'$ are close for small $t_{\text{Boom}}$, which can be seen in Figure 1.

When performing the forward process of Boomerang, it is not necessary to iteratively add noise $t_{\text{Boom}}$ times. Instead, we simply calculate the corresponding $\alpha_{t_{\text{Boom}}}$ and sample from Equation (2) once to avoid unnecessary computations. The reverse process must be done step by step however, which is where most of the computations take place, much like regular (global) sampling of diffusion models. Nonetheless, sampling with Boomerang has significantly lower computational costs than global sampling; the time required for Boomerang is approximately $\frac{t_{\text{Boom}}}{T}$ times the time for regular sampling. Moreover, we can use Boomerang to perform local sampling along with faster sampling schedules, e.g., sampling schedules that reduce sampling time by 90% (Kong & Ping, 2021) before Boomerang is applied. Pseudocode for the Boomerang algorithm is shown in Algorithm 1.

We present a quantitative analysis to measure the variability of Boomerang-generated images as $t_{\text{Boom}}$ is changed. As an expression of this variability, we consider the distribution of samples generated through the Boomerang procedure conditioned on the associated noisy input image at step $t_{\text{Boom}}$, i.e., $p_\phi(\boldsymbol{x}_0'|\boldsymbol{x}_{t_{\text{Boom}}})$. According to Bayes' rule, we relate this distribution to the distribution of noisy images at step $t_{\text{Boom}}$ of the forward process,

$$\begin{aligned} p_\phi(\boldsymbol{x}_0'|\boldsymbol{x}_{t_{\text{Boom}}}) &\propto p_\phi(\boldsymbol{x}_{t_{\text{Boom}}}|\boldsymbol{x}_0')p(\boldsymbol{x}_0') \\ &\approx q(\boldsymbol{x}_{t_{\text{Boom}}}|\boldsymbol{x}_0')p(\boldsymbol{x}_0') \\ &= \mathcal{N}\left(\sqrt{\alpha_{t_{\text{Boom}}}}\boldsymbol{x}_0', (1 - \alpha_{t_{\text{Boom}}})\boldsymbol{I}\right) p(\boldsymbol{x}_0). \end{aligned} \tag{7}$$

---

**Algorithm 1** Boomerang local sampling, given a diffusion model $f_\phi(\boldsymbol{x}, t)$

---

**Input:** $\boldsymbol{x}_0$, $t_{\text{Boom}}$, $\{\alpha_t\}_{t=1}^T$, $\{\beta_t\}_{t=1}^T$
**Output:** $\boldsymbol{x}_0'$
  $\boldsymbol{\epsilon} \leftarrow \mathcal{N}(\boldsymbol{0}, \boldsymbol{I})$
  $\boldsymbol{x}_{t_{\text{Boom}}}' \leftarrow \sqrt{\alpha_{t_{\text{Boom}}}}\boldsymbol{x}_0 + \sqrt{1 - \alpha_{t_{\text{Boom}}}}\boldsymbol{\epsilon}$
  **for** $t = t_{\text{Boom}}, ..., 1$ **do**
    **if** $t > 1$ **then**
      $\tilde{\beta}_t = \frac{1 - \alpha_{t-1}}{1 - \alpha_t}\beta_t$
      $\boldsymbol{\eta} \sim N(\boldsymbol{0}, \tilde{\beta}_t\boldsymbol{I})$
    **else**
      $\boldsymbol{\eta} = \boldsymbol{0}$
    **end if**
    $\boldsymbol{x}_{t-1}' \leftarrow f_\phi(\boldsymbol{x}_t', t) + \boldsymbol{\eta}$
  **end for**
  **return** $\boldsymbol{x}_0'$

---

The second line in the expression above assumes that by training the diffusion model via the loss function in Equation (6), the model will be able to reverse the diffusion process at each step of the process. The last line in the equation above follows from Equation (2) and the fact that $p(\boldsymbol{x}_0') = p(\boldsymbol{x}_0)$. The latter can be understood by noting that $p(\mathbf{x}_0')$ is the distribution of images generated by the diffusion model when the reverse process is initiated at step $t_{\text{Boom}}$ using noisy images obtained from $\mathbf{x}_{t_{\text{Boom}}}' = \sqrt{\alpha_{t_{\text{Boom}}}}\mathbf{x}_0 + \sqrt{1 - \alpha_{t_{\text{Boom}}}}\boldsymbol{\epsilon}$, where the original images $\mathbf{x}_0$ are drawn from $p(\mathbf{x}_0)$ and $\boldsymbol{\epsilon} \sim \mathcal{N}(\boldsymbol{0}, \mathbf{I})$. The distribution of these noisy images is equivalent to $\mathcal{N}(\sqrt{\alpha_{t_{\text{Boom}}}}\mathbf{x}_0, 1 - \alpha_{t_{\text{Boom}}}\mathbf{I})$, with $\mathbf{x}_0 \sim p(\mathbf{x}_0)$, which is equal to the forward diffusion process distribution at step $t_{\text{Boom}}$, denoted as $q(\mathbf{x}_{t_{\text{Boom}}}|\mathbf{x}_0)$ (recall Equation (2)). Given that the diffusion model is well-trained, we can expect that its output matches the original image distribution regardless of which step the reverse process in initiated, as long as the same forward diffusion process noise schedule is used. Equation (7) suggests that the density of Boomerang-generated images is proportional to the density of a Gaussian distribution with covariance $(1 - \alpha_{t_{\text{Boom}}})\boldsymbol{I}$ times the clean image density $p(\boldsymbol{x}_0)$. In other words, the resulting density will have very small values far away from the mean of the Gaussian distribution. In addition, the high probability region of $p_\phi(\boldsymbol{x}_0'|\boldsymbol{x}_{t_{\text{Boom}}})$ grows as $1 - \alpha_{t_{\text{Boom}}}$ becomes larger. This quantity monotonically increases as $t_{\text{Boom}}$ goes from one to $T$ since $\alpha_t = \prod_{i=1}^t(1 - \beta_i)$ and $\beta_i \in (0, 1)$. As a result, we expect the variability in Boomerang-generated images to increase as we run Boomerang for larger $t_{\text{Boom}}$ steps.

Since Boomerang depends on a pretrained diffusion model $f_\phi$, it does not require the user to have access to significant amount of computational resources or data. This makes Boomerang very accessible to practitioners and even everyday users who do not have specialized datasets or hardware to train a diffusion model for their specific problem. Boomerang requires that the practitioner find a diffusion model that represents the desired image manifold. With the advent of diffusion models trained on diverse datasets, such as Stable Diffusion (Rombach et al., 2022), finding such models is becoming less and less of a problem. Overall, our Boomerang method allows local sampling on image manifolds without requiring significant amounts of computational resources or data.

## 4 Application 1: Anonymization of data

Local sampling anonymizes data by replacing original data with close, but nonidentical, samples on the learned data manifold. Overparameterized (i.e., deep) models are vulnerable to membership inference attacks (Shokri et al., 2017; Tan et al., 2022), which attempt to recover potentially sensitive data (e.g., medical data, financial information, and private images) given only access to a model that was trained on said data. Boomerang local sampling enables privacy-preserving machine learning by generating data points that are similar to real (i.e., sensitive) data, yet not the same. The degree of similarity to the real data can be coarsely controlled through the $t_{\text{Boom}}$ parameter, however Boomerang can not remove specific sen-

| Class | 0 | 1 | 2 | 3 | 4 | 5 | 6 | 7 | 8 | 9 |

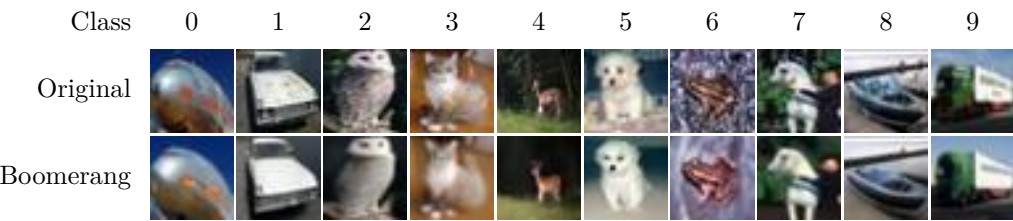

Figure 2: Using Boomerang on CIFAR-10 to change the visual features of images. These images were created with FastDPM (Kong & Ping, 2021) using $t_{\text{Boom}}/T = \frac{40}{100} = 40\%$.

sitive attributes while retaining other attributes. We prove the effectiveness of Boomerang anonymization by anonymizing various datasets, quantitatively measuring the degree of anonymity for anonymized face datasets, and successfully training classifiers on fully anonymous datasets.

### 4.1 Data and models

To show the versatility of Boomerang anonymization, we apply it to several datasets such as the LFWPeople (Huang et al., 2007), CelebA-HQ (Karras et al., 2018), CIFAR-10, CIFAR-100 (Krizhevsky et al., 2009), FFHQ (Karras et al., 2019), and ILSVRC2012 (ImageNet) (Russakovsky et al., 2015) datasets. For the ImageNet-200 experiments, we use a 200-class subset of ImageNet that we call ImageNet-200; these are the 200 classes that correspond to Tiny ImageNet (Russakovsky et al., 2015). Furthermore, to show Boomerang can be applied independent of the specific diffusion model architecture, we apply Boomerang to the Stable Diffusion (Rombach et al., 2022), Patched Diffusion (Luhman & Luhman, 2022),[1], Denoising Likelihood Score Matching (DLSM) (Chao et al., 2022)[2], and FastDPM (Kong & Ping, 2021)[3] models. We compare Boomerang-generated data with purely synthetic data from the SOTA StyleGAN-XL (Sauer et al., 2022) and DLSM models.

When generating Boomerang samples for data anonymization or augmentation, we pick $t_{\text{Boom}}$ so that the Boomerang samples look visually different than the original samples. With the FastDPM model we use $t_{\text{Boom}}/T = 40/100 = 40\%$[3]; with Patched Diffusion, we use $t_{\text{Boom}}/T = 75/250 = 30\%$; and with DLSM, we use $t_{\text{Boom}}/T = 250/1000 = 25\%$.

### 4.2 Anonymization

Boomerang can anonymize entire datasets to varying degrees controlled by the hyperparameter $t_{\text{Boom}}$, which coarsely defines the anonymization level. For example, we anonymize commonly used datasets of face images. Additionally, we anonymize natural images. Specifically, we define that a natural image $x_0$ is anonymized to $x_0'$ if the features of each image are visibly different such that an observer would guess that the two images are of different objects (note that we do not control which features are being anonymized). For each diffusion model, we pick $t_{\text{Boom}}$ so that the anonymized images are different, but not drastically different from the original dataset images. Some examples of anonymized images are shown in Figures 2 to 4.

Using Boomerang, we successfully anonymize datasets of faces, such that established facial recognition networks infer that the anonymized and original images are from different people. First, we apply Boomerang Stable Diffusion to the LFWPeople and CelebA-HQ datasets at several ratios of $t_{\text{Boom}}/T : \{0.2, 0.5, 0.7, 0.8\}$. Random samples (Figure 4) qualitatively establish that sufficiently large values of $t_{\text{Boom}}$ replace identifiable features of the original images. Meanwhile, dataset-wide evaluations via facial recognition network embedding distances show that the distributions of perceptual dissimilarity between the original and Boomerang-anonymized images shift upward as a function of $t_{\text{Boom}}$ (Figure 5 and Appendix A.1). In fact, the percentage of Boomerang-anonymized images that the facial recognition networks declare as originating from different people than the original dataset images approaches 100% as $t_{\text{Boom}}$ increases. Therefore, we qualitatively

---

[1] We use this repository for Patched Diffusion.

[2] We use this repository for DLSM.

[3] We use this repository models. It distills the original 1000 diffusion steps down to 100 via a STEP DDPM sampler.

| Class | 0 | 32 | 67 | 142 | 183 |

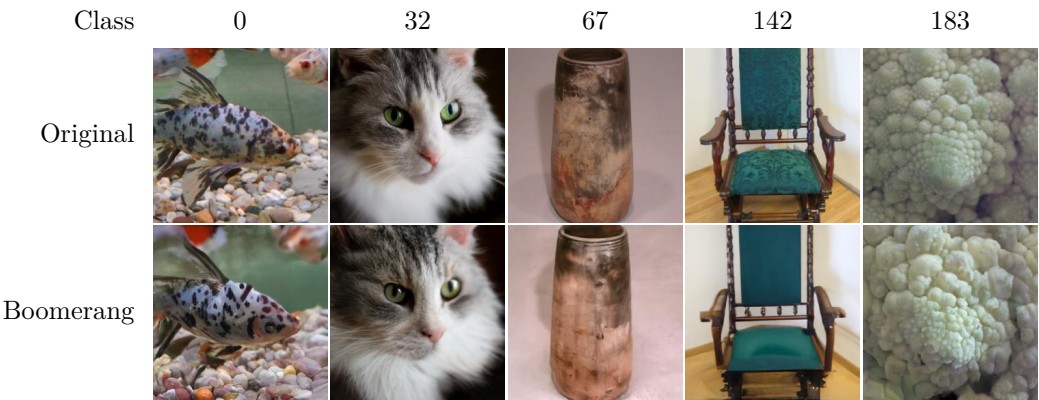

Figure 3: Using Boomerang on ImageNet-200 to change the visual features of images. These images were created with Patched Diffusion (Luhman & Luhman, 2022) using $t_{\text{Boom}}/T = 75/250 = 30\%$. The FID values for these images have been plotted in Figure 15 in the Appendix.

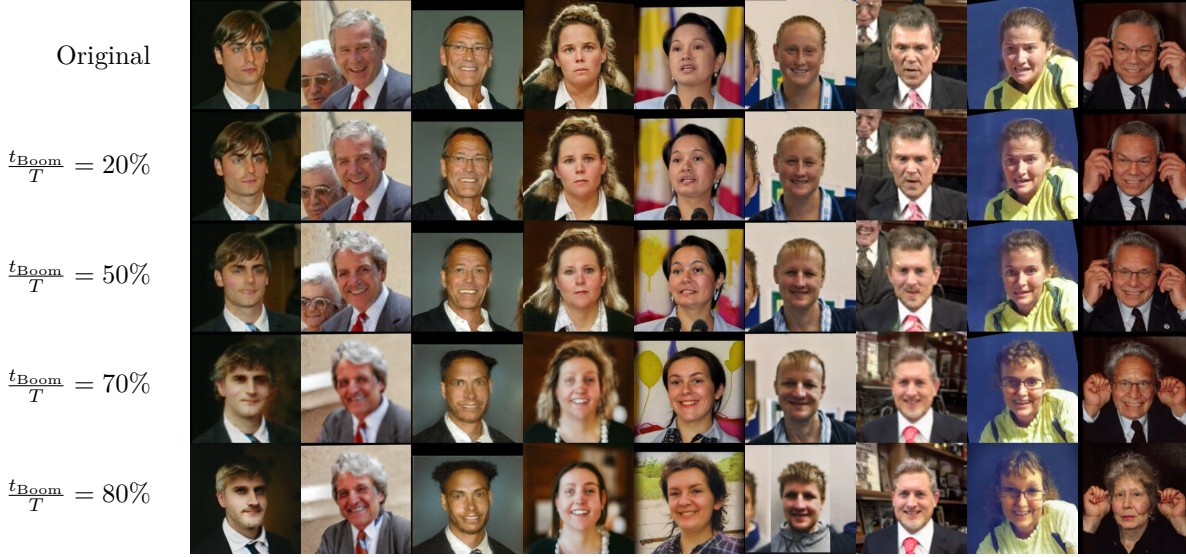

Figure 4: Nine randomly selected LFWPeople samples anonymized via Boomerang Stable Diffusion with the same random seed and with the same prompt: "picture of a person".

and quantitatively establish that Boomerang anonymization is an efficient method of anonymizing images by local sampling.

Since training on anonymous data enables privacy-preserving machine learning, we show in Table 1 that our local sampling method outperforms state-of-the-art generative models on the task of generating entirely anonymous datasets upon which to train a classifier. Specifically, we compare Boomerang anonymization against the strongest alternative: image synthesis by StyleGAN-XL (SOTA for CIFAR-10 and ImageNet) and DLSM (SOTA for CIFAR-100). We observe across all tasks that purely synthetic data, even from SOTA models, cannot reach the quality of data anonymized via Boomerang local sampling—even when the local sampling is done with diffusion models that are not SOTA. For example, on ImageNet the synthetic data produces a test accuracy of 39.8%, while the Boomerang-anonymized data produces a test accuracy of 57.8%. This phenomenon makes sense because anonymization via local sampling is in-between generating purely synthetic data and using real data. Thus, training on Boomerang-anonymized data outperforms training on synthetic data.

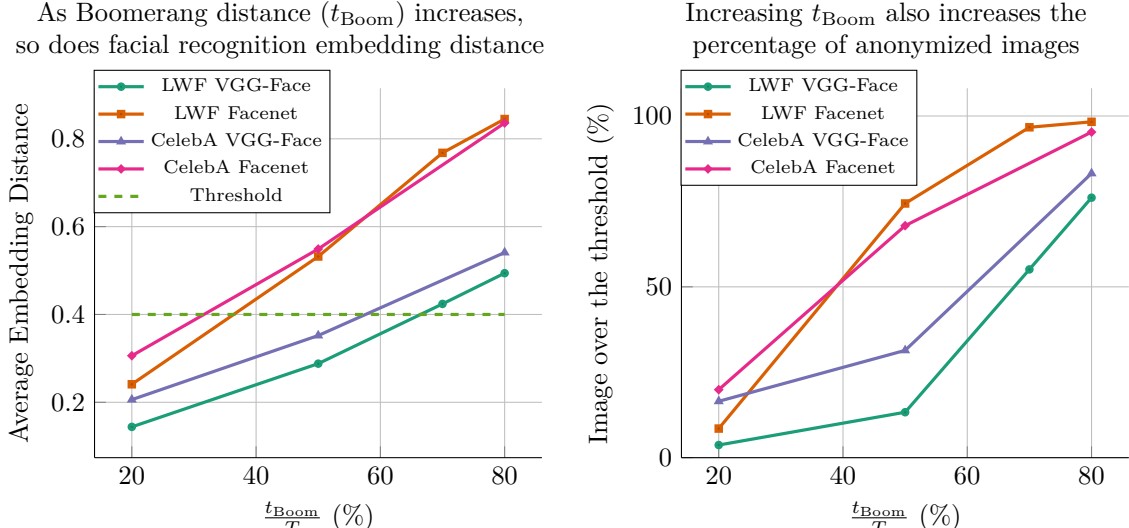

Figure 5: Facial recognition embedding distances between Boomeranged images and the original images as a function of $t_{\text{Boom}}$. We use the VGG-Face and Facenet models (Wang & Deng, 2021) to calculate the embeddings; both models have a default minimum distance threshold of 0.4 to declare that two images are of different people. The largest standard error is 0.0017 for embedding distance and 0.21% for the number of images over the threshold. Figure 10 contains the full distribution of VGG-Face embedding distances.

Table 1: With respect to training CIFAR-10, CIFAR-100, ImageNet-200, and ImageNet classifiers on anonymous data, Boomerang-anonymized data is a superior alternative to purely synthetic data from the SOTA StyleGAN-XL and DLSM models.

| Classification Task | Training data | Top-1 Test Accuracy | Top-5 Test Accuracy |
|---|---|---|---|
| CIFAR-10 | StyleGAN-XL generated data (SOTA synthetic) | 81.5% | |
| CIFAR-10 | FastDPM Boomerang data (ours) | **84.4**% | |
| CIFAR-10 | CIFAR-10 data (no anonymization) | 87.8% | |
| CIFAR-100 | DLSM generated data (SOTA synthetic) | 26.9% | |
| CIFAR-100 | DLSM Boomerang data (ours) | **55.6%** | |
| CIFAR-100 | CIFAR-100 data (no anonymization) | 62.7% | |
| ImageNet-200 | StyleGAN-XL generated data (SOTA synthetic) | 50.2% | 73.0% |
| ImageNet-200 | Patched Diffusion Boomerang data (ours) | **61.8%** | **83.4%** |
| ImageNet-200 | ImageNet-200 data (no anonymization) | 66.6% | 85.6% |
| ImageNet | StyleGAN-XL generated data (SOTA synthetic) | 39.8% | 62.1% |
| ImageNet | Patched Diffusion Boomerang data (ours) | **57.8%** | **81.3%** |
| ImageNet | ImageNet data (no anonymization) | 63.3% | 85.3% |

In conclusion, we have established that data anonymization via Boomerang can operate on entire datasets, successfully anonymize facial images, and is a superior alternative to generating purely synthetic data for downstream tasks such as image classification.

Table 2: Boomerang-generated data for data augmentation increases test accuracy of CIFAR-100, ImageNet-200, and ImageNet classification tasks. The purely sythetic data augmentation was done using state-of-the-art (SOTA) models: Denoising Likelihood Score Matching (DLSM) (Chao et al., 2022) and StyleGAN-XL (Sauer et al., 2022). For the Boomerang data, we used DLSM and Patched Diffusion (Luhman & Luhman, 2022).

| Classification Task | Training data | Top-1 Test Accuracy | Top-5 Test Accuracy |
|---|---|---|---|
| CIFAR-100 | CIFAR-100 only (no data augmentation) | 62.7% | |
| CIFAR-100 | CIFAR-100 + DLSM DA (SOTA synthetic) | 56.0% | |
| CIFAR-100 | CIFAR-100 + DLSM Boomerang DA (ours) | **63.6%** | |
| ImageNet-200 | ImageNet-200 only (no data augmentation) | 66.6% | 85.6% |
| ImageNet-200 | ImageNet-200 + StyleGAN-XL DA (SOTA synthetic) | 65.8% | 85.5% |
| ImageNet-200 | ImageNet-200 + Patched Diffusion Boomerang DA (ours) | **70.5%** | **88.3%** |
| ImageNet | ImageNet only (no data augmentation) | 63.3% | 85.3% |
| ImageNet | ImageNet + StyleGAN-XL DA (SOTA synthetic) | 63.3% | 85.3% |
| ImageNet | ImageNet + Patched Diffusion Boomerang DA (ours) | **64.4%** | **86.0%** |

## 5 Application 2: Data augmentation

Data augmentation for image classification is essentially done by sampling points on the image manifold near the training data. Typical augmentation techniques include using random image flips, rotations, and crops, which exploit symmetry and translation invariance properties of images in order to create new data from the training set. Although there are many techniques for data augmentation, most involve modifying the training data in ways which make the new data resemble the original data while still being different, i.e., they attempt to perform local sampling on the image manifold. Since Boomerang can also locally sample on manifolds, we investigate if using Boomerang is beneficial for data augmentation on classification tasks.

For our data augmentation experiments, we pick $t_{\mathrm{Boom}}$ to be large enough to produce visual differences between the original dataset and the Boomerang-generated dataset, as shown in Figures 2 and 3; as discussed Section 4.1. Due to the intrinsic computational costs of diffusion models, we generate the augmented data before training instead of on-the-fly generation during training. We then randomly choose to use the training data or the Boomerang-generated data with probability 0.5 at each epoch. We use ResNet-18 (He et al., 2016) for our experiments. We use the same models and datasets as described in Section 4.1.

To demonstrate the impact of data augmentation using Boomerang, we begin by comparing how it enhances accuracy in classification tasks as compared to the case of no data augmentation. We find that using Boomerang for data augmentation increases generalization performance over not using data augmentation at all for a wide range of datasets and classifier architectures. As shown on Table 2, using Boomerang data augmentation on CIFAR-100, ImageNet-200, and ImageNet classification tasks increases test accuracy from 62.7% to 63.6%, from 66.6% to 70.5%, and from 63.3% to 64.4%, respectively. Applying Boomerang data augmentation requires only selecting a single hyperparameter, namely $t_{\mathrm{Boom}}$, and it does not make any explicit assumptions about data invariances, making it a versatile and general-purpose data augmentation tool. Additionally, Boomerang data augmentation can be combined with other data augmentation techniques, e.g., flips, crops, and rotations.

We also observe that using Boomerang for data augmentation increases generalization performance over using purely synthetic data augmentation. Specifically, we see that using purely synthetic data augmentation does not appear to help generalization performance at all. The generalization performance is actually *lower than or equal to the baseline* for CIFAR-100, ImageNet-200, and ImageNet classifications tasks with SOTA synthetic data augmentation (see Table 2). This is somewhat surprising because the models that we used for the Boomerang data augmentation are not SOTA, with the exception of the DLSM model. Meanwhile, the models that we use for the purely synthetic data augmentation are SOTA and thus we compare to

the hardest setting possible. For example, the reported FID (metric of image quality, lower is better) for the Patched Diffusion model we use for Boomerang on ImageNet is 8.57 whereas it is 2.3 for StyleGAN-XL Luhman & Luhman (2022), which is used for the synthetic data augmentation. Therefore, Boomerang data augmentation seems to be more beneficial than using purely synthetic data augmentation even if Boomerang is done with a lower performing model.

In conclusion, we showed that Boomerang data augmentation can be used to increase generalization performance over no augmentation and that it even beats synthetic data augmentations using SOTA models.

## 6 Application 3: Perceptual resolution enhancement for low-resolution images

As the final application of Boomerang local sampling, we propose perceptual resolution enhancement (PRE) for images. This process aims to upsample low-resolution images with a focus on perceptual quality even if traditional handcrafted quality measures such as peak signal-to-noise ratio (PSNR) are low. We begin by framing PRE as a local sampling problem, followed by two proposed perceptual resolution enhancement PRE methods using Boomerang.

### 6.1 Perceptual resolution enhancement via local sampling

Downsampled or otherwise corrupted images belong to a different distribution than realistic or natural images. For example, the dimension of a ground-truth image $x_{\text{true}} \in \mathbb{R}^n$ is higher than that of its downsampled image $x_{\text{ds}} \in \mathbb{R}^d$. We frame PRE as a local sampling approach to restore $x_{\text{ds}}$ through: (1) obtaining a rough approximation $x_{\text{up}} \in \mathbb{R}^n$ of the ground-truth high-resolution image using standard resolution enhancement techniques, and (2) restoring perceptual quality via applying Boomerang to $x_{\text{up}}$. By locally sampling around $x_{\text{up}} \approx x_{\text{true}}$, Boomerang produces a point on the manifold of high-resolution images near the ground-truth image $x_{\text{true}}$. The goal of PRE is to improve the perceptual quality of a corrupted or low-dimensional image $x_{\text{ds}}$ without being forced to match $x_{\text{ds}}$ at every pixel: i.e., unlike traditional super-resolution, PRE is allowed to modify the pixel values of the downsampled image instead of just interpolating between them.

### 6.2 Vanilla Boomerang perceptual resolution enhancement

We perform Boomerang PRE by:

1. Bringing the corrupted image into the ambient space of the desired result using some off-the-shelf interpolation method (cubic, nearest-neighbor, etc). In all our experiments, we used linear interpolation.

2. Selecting a $t_{\text{Boom}}$ corresponding to the desired "radius" of the search space. As $\frac{t_{\text{Boom}}}{T} \to 1$, we move from locally sampling around a given point to sampling globally on the learned manifold.

3. Performing Boomerang as described in Section 3.

The resulting image is an enhanced version of the input image in the neighborhood of the ground-truth image. The parameter $t_{\text{Boom}}$ corresponds to the strength of PRE applied, with larger values more appropriate for more corrupted images.

While others such as Saharia et al. (2022b) and Rombach et al. (2022) have used diffusion models for image enhancement (including super-resolution), Boomerang has two key advantages to existing diffusion-model-based methods. The first is that Boomerang's local sampling keeps the output "close" to the input on the image manifold, i.e., we implicitly account for the geodesics of the manifold instead of merely optimizing a metric in the Euclidean space. The second is that adjusting $t_{\text{Boom}}$ allows easy "tuning" of the PRE strength, controlling the resulting trade-off between image fidelity and perceptual quality. This means that *the same pretrained network* can perform perceptual enhancement on images with different levels of degradation or downsampling: one can choose a larger value of $t_{\text{Boom}}$ to fill in more details of the final image. Furthermore, by varying $t_{\text{Boom}}$ for different passes of the same input image, one can generate multiple images, each with a

different detail/variance trade-off. This can be seen in Figure 11 in Appendix A.2, wherein more aggressive enhancement improves clarity at the cost of distance from the ground-truth image. Empirical tests showed that setting $t_{\text{Boom}} \approx 100$ on the Patched Diffusion Model (out of $T = 250$) produced a good balance between sharpness and the features of the ground-truth image, as seen in Appendix A.2. Finally, the same image can be passed through Boomerang multiple times with a fixed $t_{\text{Boom}}$ to generate different candidate images.

Our subsequent PRE experiments with Boomerang all use the Patch Diffusion model (Luhman & Luhman, 2022). With Stable Diffusion (Rombach et al., 2022), empirical results showed that detail was not added to the low-resolution images except for large $t_{\text{Boom}}$, where generated images no longer strongly resembled the ground truth images. We believe this is because noise is added in the latent space with Stable Diffusion (as opposed to in the image space with Patched Diffusion). We conclude that the noise impacts which kinds of inverse problems Boomerang will be effective on.

Since Boomerang PRE emphasizes perceptual quality, we evaluate its performance on metrics correlating with human perceptual quality, such as Fréchet Inception Distance (FID) and Learned Perceptual Image Patch Similarity (LPIPS) (Heusel et al., 2017; Zhang et al., 2018). FID measures the distance (lower is better) between the distributions of "real" images and generated images. LPIPS, on the other hand, compares deep embedding of corresponding patches between two images, and correlates well with human visual perception. We follow (Zhang et al., 2018)'s recommendation and use AlexNet as the backend for LPIPS. Unlike FID, LPIPS is an image-to-image metric. We compare Boomerang PRE with (1) classical super-resolution methods such as nearest interpolation, linear interpolation, and cubic interpolation; and (2) the Deep Image Prior (DIP) (Ulyanov et al., 2020), an *untrained* method that performs well on super-resolution tasks. We chose not to compare Boomerang to deep learning methods explicitly trained to do image enhancement or super-resolution, as Boomerang and DIP do not require training nor fine-tuning.

Boomerang provides superior performance to all competing methods on perceptual metrics such as LPIPS and FID, as seen in Table 3. Furthermore, the hyperparameter $t_{\text{Boom}}$ provides a unique trade-off between data fidelity and perceptual quality, enabling cost-effective and controllable generation of perceptually enhanced images. We present the Boomerang perceptual enhancement results in Table 3.

Another benefit of Boomerang is its flexibility: Boomerang can perform variable perceptual image enhancement at various degrees of degradation using the same pretrained diffusion model. This is shown in Figure 13, where we enhance images downsampled by 4x, 8x and even 16x by simply varying $t_{\text{Boom}}$. Although methods like DIP and linear interpolation are also flexible to the extent they are applicable for images downsampled by different factors, many superresolution methods are not. In particular, many deep learning superresolution methods require separate training or fine-tuning depending on the desired fidelity, which requires more compute and explicit model selection. In our examples, which used $1024 \times 1024$ images, we found $t_{\text{Boom}} = 50$ worked well with images downsampled by 4x while $t_{\text{Boom}} = 100$ worked best with images downsampled by 8x. Our intuition suggests that a larger value of $t_{\text{Boom}}$ is more appropriate for more degraded images, and our empirical results are in agreement.

Furthermore, Boomerang PRE is relatively fast and computationally efficient. On $1024 \times 1024$ images, Boomerang takes 0.932 minutes with a standard deviation of $0.0321^4$ minutes per image. On the other hand, DIP takes about 36.426 minutes with a standard of 0.867 minutes per image. All of these statistics were calculated using random 50 images. In the time it takes to perform PRE on one image with DIP, 39 images can be enhanced with Boomerang. Additionally, Boomerang benefits from diffusion acceleration techniques like batch processing and distillation, whereas DIP cannot because each image is its own optimization problem. Although classical methods, such as linear interpolation, are much faster than Boomerang, they are not perceptually pleasing and were not specifically designed to do data-driven perceptual enhancement.

### 6.3 Cascaded Boomerang image enhancement

As $t_{\text{Boom}}$ is increased, the variance of the generated images dramatically increases due to diffusion models' stochastic nature (noise is added to data at each step $t$, see Algorithm 1). As a result, increasing $t_{\text{Boom}}$ eventually causes the generated images to vary so much that they no longer resemble the input image at all

---

[4]These times are reported for a single Nvidia GeForce GTX Titan X GPU

Table 3: FID and LPIPS scores of different super-resolution methods (lower is better). Each was calculated from 5000 images from the FFHQ dataset (Karras et al., 2019). Since LPIPS is an image-to-image metric, these values are the average scores across the individual images. FID scores around 5 are within the common range between test and training splits of the same dataset. For Cascaded results, the number of cascades were selected based on the best images as decided by a human observer who chose the image that had the best perceptual quality.

| Image Enhancement Method | LPIPS Score | FID |
|---|---|---|
| Nearest Interpolation | 0.550 | 18.82 |
| Linear Interpolation | 0.449 | 20.98 |
| Cubic Interpolation | 0.443 | 16.60 |
| Deep Image Prior | 0.353 | 7.14 |
| Boomerang with $t_{\text{Boom}} = 50$ | 0.341 | 8.93 |
| Boomerang with $t_{\text{Boom}} = 100$ | **0.338** | 5.10 |
| Boomerang with $t_{\text{Boom}} = 150$ | 0.394 | 5.19 |
| Boomerang Cascaded, $t_{\text{Boom}} = 50$ | 0.351 | **4.47** |

(i.e., we move from local sampling to global sampling). Even for modest values of $t_{\text{Boom}}$, the large variance of added noise causes repeated PRE attempts to differ significantly. In this section, we propose a simple method to keep variance of the generated image manageable.

Cascaded Boomerang describes repeated passes of a corrupted image through a diffusion network with a small value of $t_{\text{Boom}}$, as opposed to passing an image through Boomerang once with a large value of $t_{\text{Boom}}$. If we denote the Boomerang PRE method in the previous section on an input image $\mathbf{x}_{\text{ds}}$ to generate $\mathbf{x}_0 = B_{f_\phi}(\mathbf{x}_{\text{ds}})$, the method described here would be $\mathbf{x}_{\text{cascade}} = B_{f_\phi}(B_{f_\phi}(\ldots(B_{f_\phi}(\mathbf{x}_{\text{ds}}))))$. We designate $n_{\text{cascade}}$ as the number of times we repeat Boomerang on the intermediate result. In addition to stabilizing independent image enhancement attempts, the cascade method allows users to iteratively choose the desired PRE detail—simply stop repeating the cascade process once the desired clarity is achieved. An example of cascaded perceptual enhancement with Boomerang is shown in Figure 6, with more examples shown in Figure 12 in Appendix A.2.

The cascade method achieved higher FID scores (worse) but lower LPIPS score (better) than vanilla Boomerang PRE. Since FID score calculates distances between distributions, and cascaded Bboomerang has a stabilizing effect on the resulting images, this is consistent with our observations. Despite higher LPIPS score, we thought the subjective quality of the cascaded Boomerang PRE images were superior to the vanilla generated images. The cascaded Boomerang method allows for progressive PRE and multiple candidate generation beyond the vanilla Boomerang method.

Boomerang is a quick and efficient method of performing perceptual resolution enhancement with pretrained diffusion models. As we have shown, Boomerang allows the user to easily adjust the strength of the enhancement by varying $t_{\text{Boom}}$. The same procedure and pretrained network can thus be used to perform perceptual enhancement at different strengths and for images at any dimension smaller than the output dimension of the diffusion network (we used images with dimensions $1024 \times 1024$): for larger scaling factors, simply increase $t_{\text{Boom}}$ or cascade the result until one achieves the desired fidelity. This also avoids the issue of needing to train different networks depending on the magnitude of enhancement desired or for a given factor of downsampling. This reflects the real-world reality that we often want to "enhance" images that may not conform to a specific scale factor or dimensionality. As with the previous applications, Boomerang's generated images look realistic yet Boomerang requires no additional fine-tuning or training.

## 7 Related work

The method that has the closest connection to our work—in terms of the underlying algorithmic procedure involving the reverse process—is SDEdit (Meng et al., 2022). This method uses pretrained unconditional diffusion models to edit or generate photorealistic images from sketches or non-natural images. While

Initial 8x downsampled Image $n_{\text{cascade}} = 5$

$n_{\text{cascade}} = 8$ Ground Truth

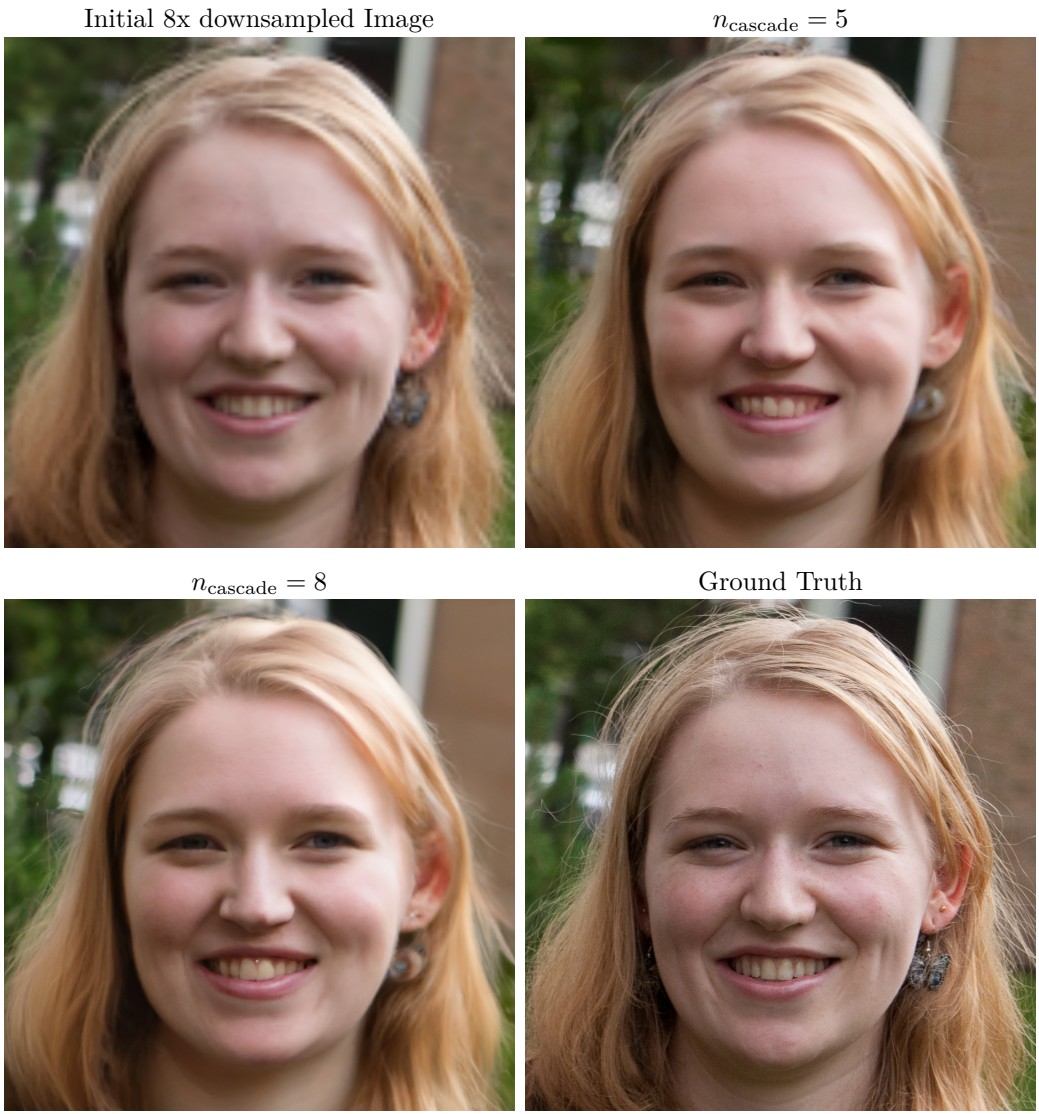

Figure 6: Cascaded Boomerang perceptual resolution enhancement with $t_{Boomerang} = 50$.

SDEdit, like Boomerang, utilizes the distinct characteristics of the reverse diffusion process to accomplish its goal, it differs fundamentally from Boomerang in that we focus on local sampling, as opposed to image editing. Specifically, we start with a natural image and generate nearby natural images on the learned manifold, and we show its applicability to data anonymization, data augmentation, and image perceptual quality enhancement. In contrast, SDEdit lacks the ability for local sampling as it requires an input sketch to generate a variation of the original natural image. Additionally, the SDEdit algorithm executes the complete forward and reverse process with an altered noise schedule, whereas Boomerang solely carries out a partial forward and reverse process without modifying the noise schedule, which makes Boomerang more computationally efficient. Inspired by SDEdit, Haque et al. (2023) employ an image-conditioned diffusion model to edit scenes generated using NeRF (Mildenhall et al., 2021). The method shares the concept of starting the diffusion reverse process from an intermediate step, but it differs from our approach in that it carries out unconditioned local sampling instead of relying on a task-specific conditional diffusion model.

In an effort to reduce the sampling cost in diffusion models, Zheng et al. (2023) proposed initiating the reverse diffusion process from an intermediate step rather than from pure Gaussian noise. To generate

samples from the diffusion model, the authors suggest utilizing another generative model, such as a VAE, to learn the distribution of noisy data at the intermediate step and employ the samples provided by this model for the reverse process. The methodology described differs from our approach as it does not involve local sampling. Instead, the authors employ a partial reverse diffusion process to expedite the sampling procedure. In their work, Chung et al. (2022) aim to decrease the computational costs of solving inverse problems that are implicitly regularized by diffusion models. To achieve this, the authors propose an iterative scheme that involves updating the estimate by taking a gradient step using a data fidelity loss function, followed by projecting it into the range of a diffusion model. To expedite the projection, the authors run the reverse process from an intermediate noisy estimate obtained by partially forward diffusing the current estimate. In contrast, our approach for enhancing image perceptual quality differs from Chung et al. (2022) in that we compute the gradient step only once at the beginning, followed by projecting it onto to the range of the diffusion model. This allows us to perform local sampling without modifying the reverse process, while Chung et al. (2022) alters the reverse process, making it conditional.

Other related work modifies the reverse diffusion processes to perform data anonymization, augmentation, or upsampling. Recently, Klemp et al. (2023) proposed a technique for dataset anonymization utilizing pretrained diffusion models. Specifically, they remove sensitive components from an image and then use a pretrained Stable Diffusion inpainting model to fill in the removed details. In comparison to their approach, our proposed method is more controllable (through the use of parameter $t_{\text{Boom}}$), requires fewer iterations to execute (especially for low values of $t_{\text{Boom}}$), and does not require the diffusion model to be specifically trained for inpainting purposes. Trabucco et al. (2023) present a novel data augmentation methodology that utilizes a pretrained text-to-image diffusion model. While Boomerang is applicable to any generic diffusion model, their approach specifically necessitates the use of a text-to-image model, and importantly, requires textual input to generate augmented images Kawar et al. (2022) propose a novel approach that leverages pretrained unconditional diffusion models to remove JPEG artifacts. Specifically, their methodology involves modifying the reverse process by conditioning it on the JPEG-compressed image. In another related work, Lugmayr et al. (2022) alter the reverse process to condition it on an image with occlusions to perform inpainting. Both of these methods for JPEG artifact removal and image inpainting utilize a partial reverse diffusion process; however, they modify the reverse process, whereas Boomerang does not make any modifications. The image editing methodology proposed by Ackermann & Li (2022) presents a multi-stage upscaling process suitable for high-resolution images using the Blended Diffusion model (Avrahami et al., 2022). Specifically, this image editing approach involves passing a low-resolution image through an off-the-shelf super-resolution model, followed by the addition of noise and reverse diffusion from an intermediate stage using the text-guided Blended diffusion model (Avrahami et al., 2022). While the usage of a partial reverse diffusion process is shared with Boomerang, this method is fundamentally different in that it aims to perform image editing instead of local sampling.

Finally, the recently introduced consistency models (Song et al., 2023) aim to reduce the computational complexity of diffusion models during inference. In their few-step method, a network is trained to perform the reverse process using a significantly coarse time discretization. The last step in the few-step method, which maps an intermediate noisy image to the final image manifold, is, in fact, an instance of Boomerang. In addition, the few-step method provides a means to trade off compute for sample quality. While this resembles the trade-off that we describe in our Boomerang-based method for perceptual image quality enhancement, in which $t_{\text{BOOM}}$ trades off compute and data fidelity for perceptual quality, consistency models are essentially a model distillation approach and do not perform local sampling.

## 8 Conclusions

We have introduced the Boomerang algorithm, which is a straightforward and computationally efficient method for performing local sampling on an image manifold using pretrained diffusion models. Given an image from the manifold, Boomerang generates images that are "close" to the original image by reversing the diffusion process starting from an intermediate diffusion step initialized by the diffused original image. The choice of intermediate diffusion step $t_{\text{Boom}}$ determines the degree of locality in Boomerang sampling, serving as a tuning parameter that can be adjusted based on the specific requirements of the downstream application. Boomerang can be run on a single GPU, without any re-training, modifications to the model,

or change in the reverse diffusion process. We showed the applicability of Boomerang on various tasks, such as anonymization, data augmentation, and perceptual enhancement. Future works include continued experiments of its efficacy for data augmentation, as well as applying the Boomerang algorithm to different domains of data, such as audio and text. Finally, recent work has shown that diffusion models can work with non-stochastic transforms instead of additive Gaussian noise (Bansal et al., 2022; Daras et al., 2022), and evaluating the Boomerang algorithms with such diffusion models would provide further insight into the nature of local sampling using diffusion models.

## Acknowledgement

This work was supported by NSF grants CCF-1911094, IIS-1838177, and IIS-1730574; ONR grants N00014-18-12571, N00014-20-1-2534, and MURI N00014-20-1-2787; AFOSR grant FA9550-22-1-0060; DOE grant DE-SC0020345; and a Vannevar Bush Faculty Fellowship, ONR grant N00014-18-1-2047.

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

# A  Appendix

## A.1  Boomerang-generated images via the Stable Diffusion model

Here we present additional images created via the Boomerang method that indicate the evolution of the predicted image as we increase $t_{\text{Boom}}$. These images are generated via the pretrained Stable Diffusion model (Rombach et al., 2022) where instead of adding noise to the image space during the forward process it is added in the latent space. Figures 7 to 9 showcase this where the images on the bottom row show noisy latent variables whereas the ones on the top row indicate the Boomerang predictions with increasing amounts of added noise from left to right, except for the rightmost image, which is created by using an alternate prompt.

## A.2  Vanilla Boomerang super-resolution

Here we present the result of Boomerang perceptual resoltion enhancement as described in Section 6.2. Figure 11 illustrates the results for image perceptual enhancement. The top-left image in this figure shows the low-resolution image, and the top-right and bottom-left images are the result of vanilla Boomerang perceptual enhancement with $t_{\text{Boom}} = 100$ and $t_{\text{Boom}} = 150$, respectively. When compared with the high-resolution image in the bottom-right corner of Figure 11 we observe that the resulting image with $t_{\text{Boom}} = 100$ is plausible while the result with $t_{\text{Boom}} = 150$ seems high-resolution, but is inconsistent compared to the ground-truth image.

# B  Trade-off between accuracy and Boomerang distance in data augmentation

The observed benefits of Boomerang in data augmentation presented in Section 5 are for specific values of $\frac{t_{\text{Boom}}}{T}$. One can observe that there would be no benefit in picking $\frac{t_{\text{Boom}}}{T} = 0\%$ because then no modification of the training data is performed and hence the augmented data will be the same as the non-augmented data, i.e., the training set. On the other hand, if we use $\frac{t_{\text{Boom}}}{T} = 100\%$, we are essentially just sampling purely synthetic images from the diffusion model and using that for data augmentation. As shown in Section 5, there exists a value of $\frac{t_{\text{Boom}}}{T}$ which is better than both of these extremes. In Figure 14 we show accuracy on the CIFAR-100 classification task for intermediate values of $\frac{t_{\text{Boom}}}{T}$, demonstrating a tradeoff between accuracy and $\frac{t_{\text{Boom}}}{T}$.

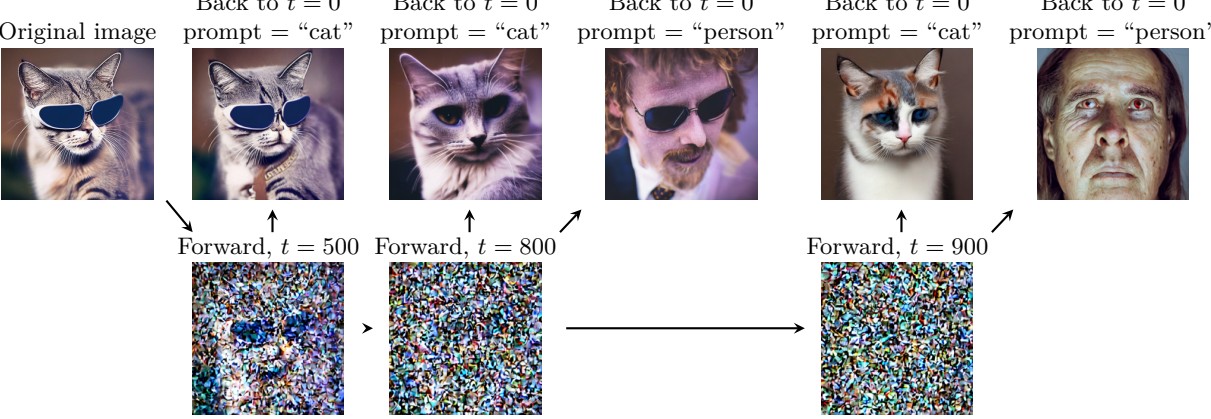

Figure 7: The Boomerang method using Stable Diffusion ($T = 1000$), as in Figure 1, with an image of a cat.

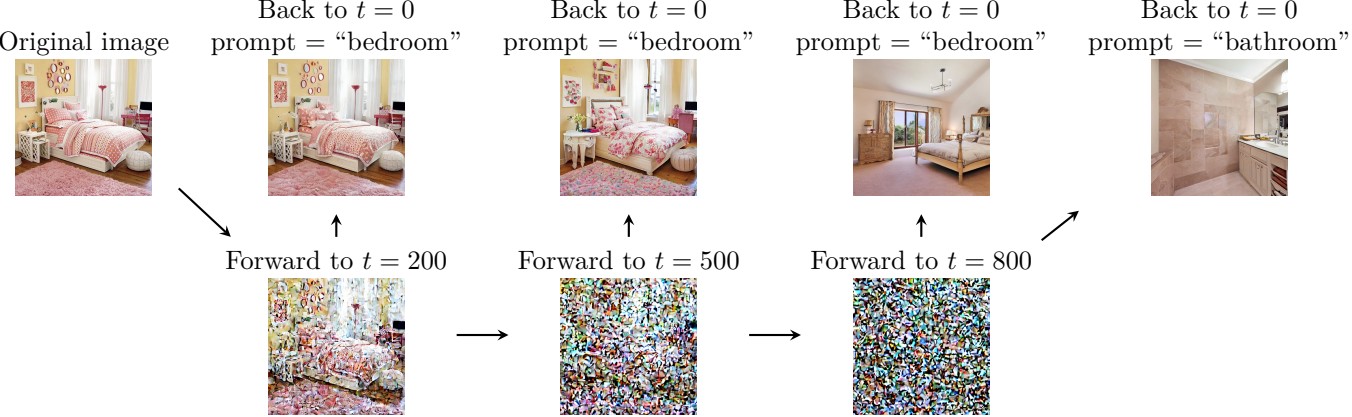

Figure 8: The Boomerang method using Stable Diffusion ($T = 1000$), as in Figure 1, with an image of a bedroom.

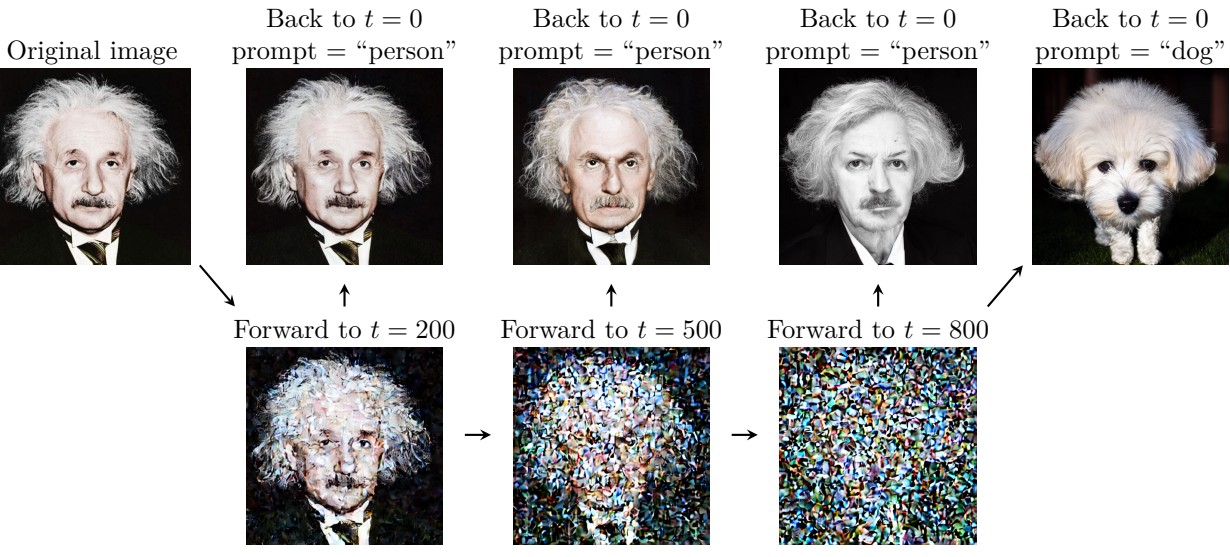

Figure 9: The Boomerang method using Stable Diffusion ($T = 1000$), as in Figure 1, with an image of Albert Einstein.

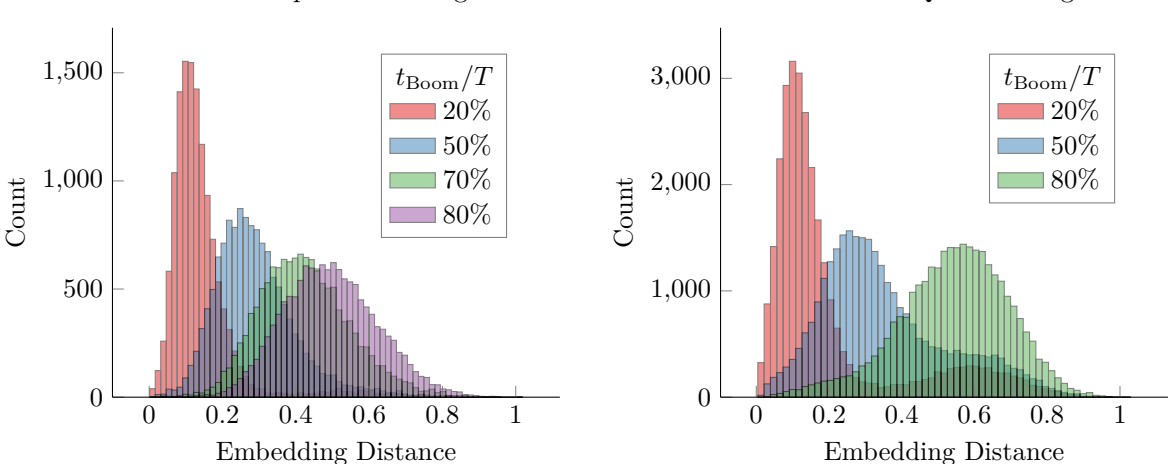

Figure 10: Distribution of facial recognition embedding distances from VGG-Face.

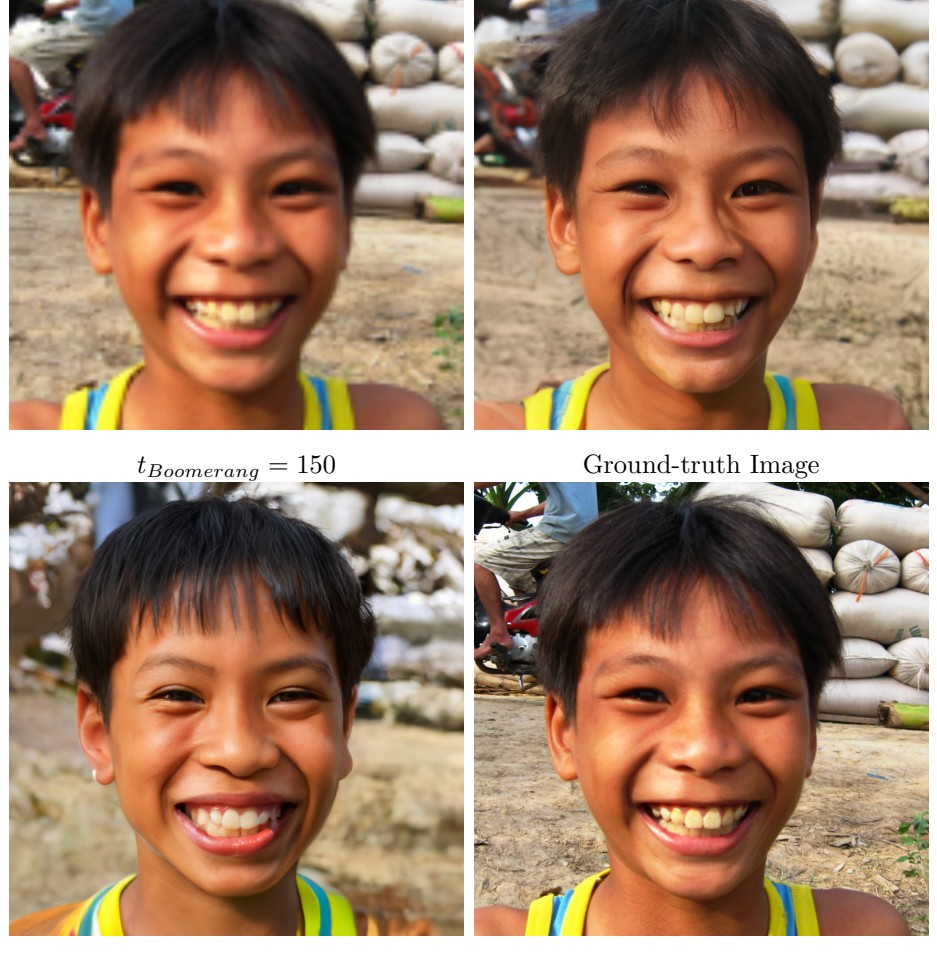

Figure 11: Boomerang perceptual resolution enhancement for different values of $t_{\text{Boom}}$.

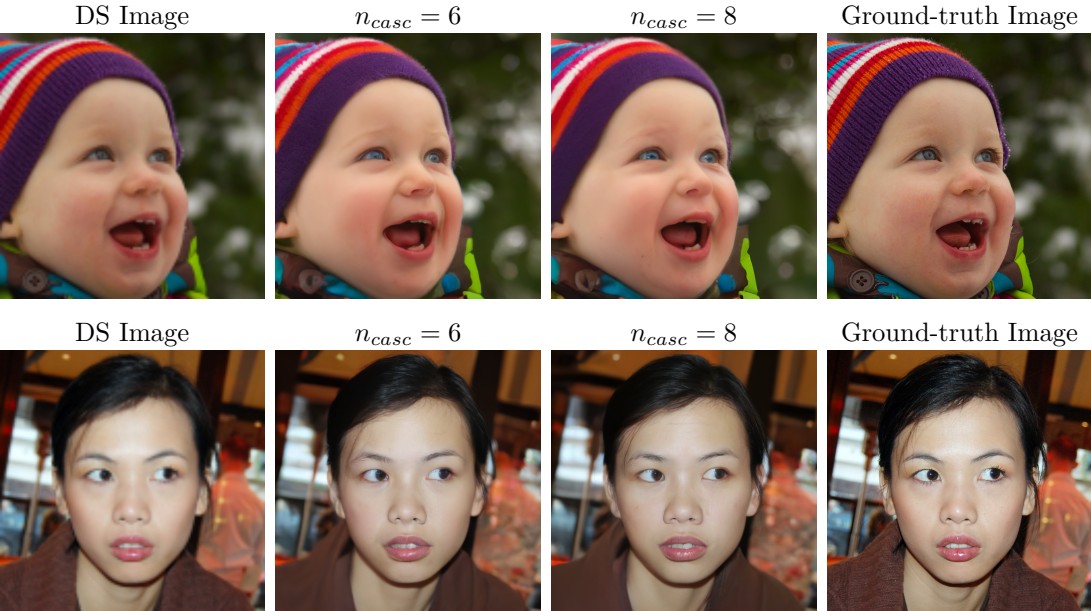

Figure 12: Cascaded Boomerang perceptual resolution enhancement. For the top row, notice how the best quality image is seen after 6 cascade steps, with $t_{Boomerang} = 50$. After 8 cascades, details such as the teeth begin to be removed. On the bottom row, however, $n_{casc} = 8$ provides better results and more detail compared to previous steps. This shows the value in introducing the cascade method: different images may have better results with more cascade steps than others.

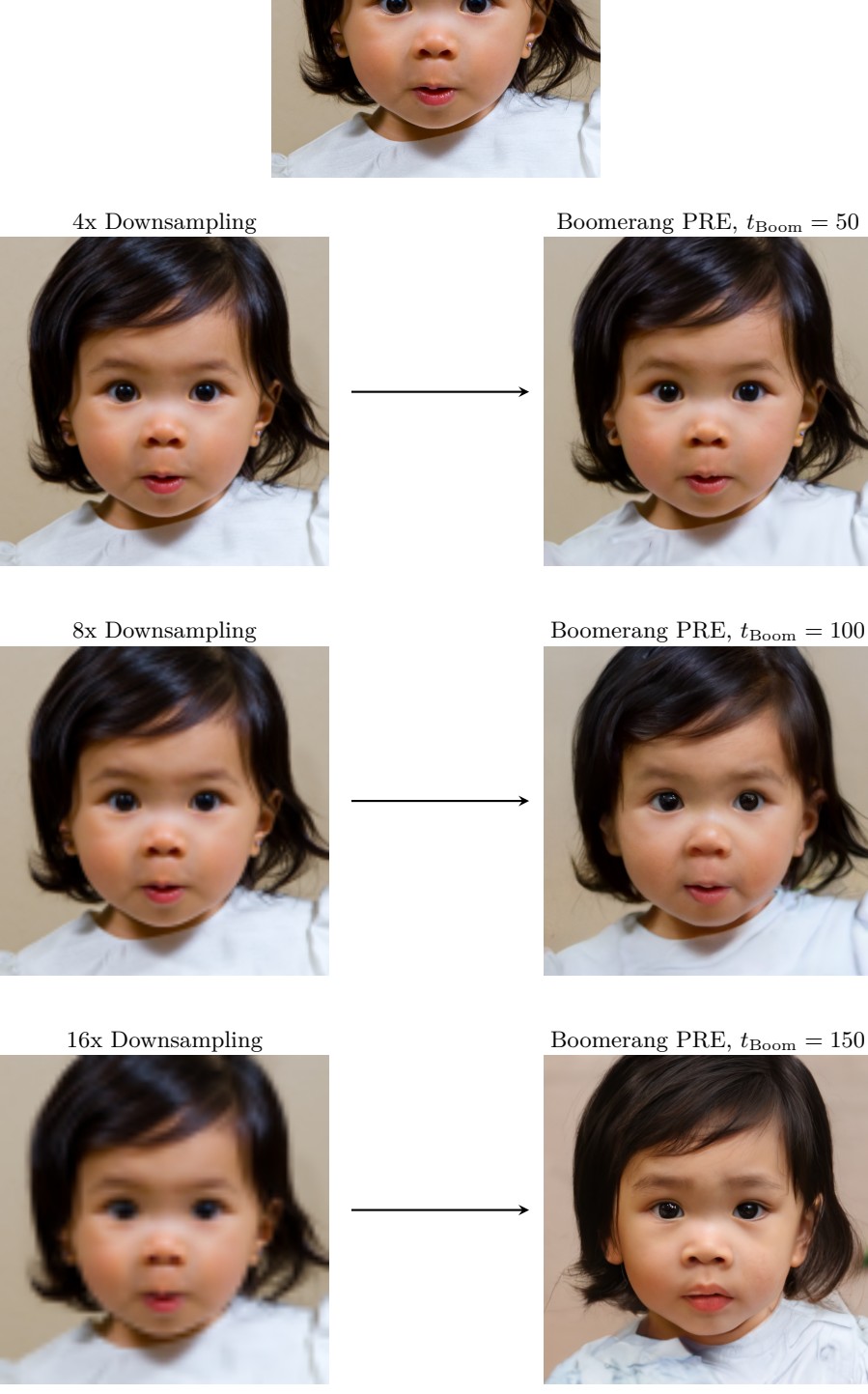

Figure 13: Perceptual resolution enhancement with Boomerang. Here, we show that by adjusting $t_{\mathrm{Boom}}$, we can easily control the strength of the perceptual resolution enhancement applied, where larger values are more appropriate for lower resolution images.

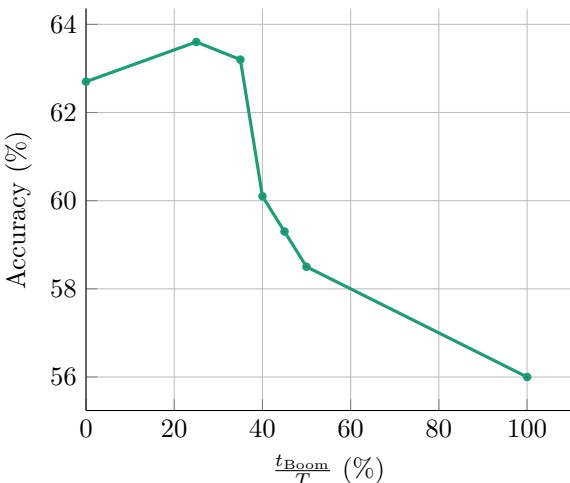

Figure 14: The trade-off between accuracy and $\frac{t_{\text{Boom}}}{T}$ has a maximum beneficial peak somewhere between 0% and 100% data augmentation. At 0% we have essentially no data augmentation because Boomerang does not modify the augmented training data. At 100% we augment with purely synthetic data, which is typically has lower quality than training data.

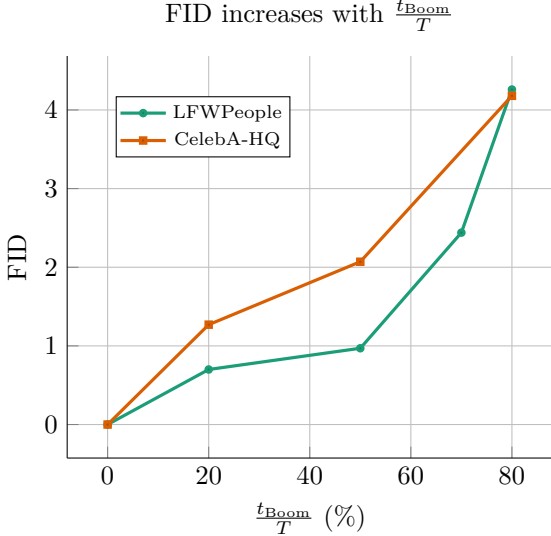

Figure 15: Using Boomerang with Stable Diffusion on the LFWPeople and CelebA-HQ datasets results in larger FID values for larger values of $\frac{t_{\text{Boom}}}{T}$. This is in part because real images are better than synthetic, even if one uses an excellent diffusion model such as Stable Diffusion.

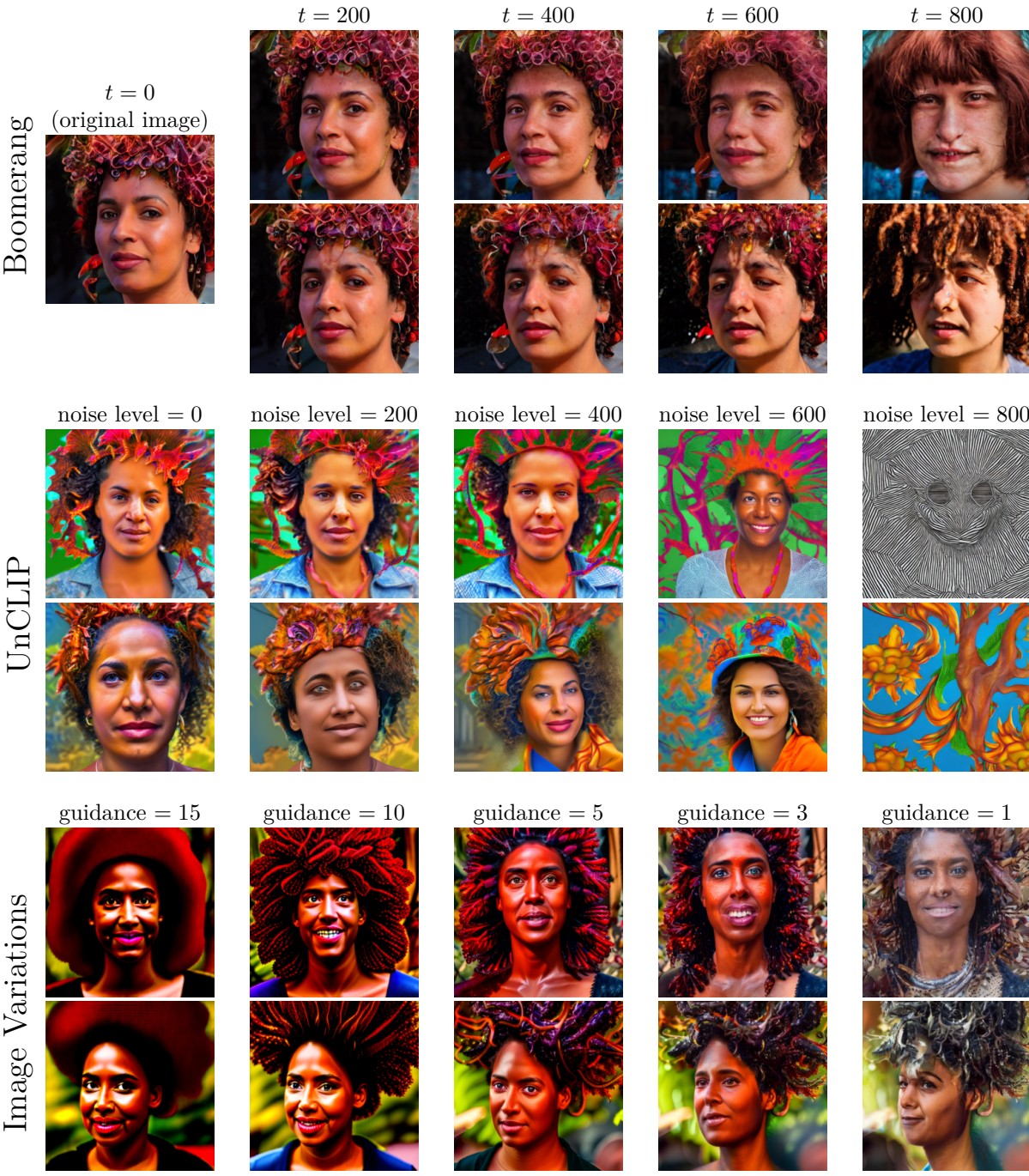

Figure 16: Here we use the same original image from Figure 1 to compare Boomerang local sampling (top) with UnCLIP (https://huggingface.co/stabilityai/stable-diffusion-2-1-unclip) and Image Variations (https://huggingface.co/lambdalabs/sd-image-variations-diffusers), which are Stable Diffusion models fine-tuned to accept CLIP guidance to produce variations of an input image. The same two random seeds are used across each method. For Boomerang and UnCLIP, we used the default guidance of 7.5 and the prompt "picture of a person". UnCLIP has a noise level parameter that behaves similarly to $t_{\text{Boom}}$. Image Variations cannot accept prompts and thus only has guidance as its input parameter. Unlike either CLIP guidance method, Boomerang can smoothly interpolate between locally sampling very closely and very distantly from the original data point while maintaining consistent image quality. For example, notice that neither CLIP guidance method can recover the original image.

