# OpenReview forum: "Boomerang: Local sampling on image manifolds using diffusion models"
_TMLR — Accepted by TMLR_

### Review · Reviewer_T5Zh · 2023-07-16

**Summary Of Contributions:**

The authors present Boomerang: a method for locally sampling from the data manifold using pre-trained diffusion models. More specifically, local sampling is achieved by noising the input data using the diffusion forward process and using the noisy sample as an initialization for the reverse process to generate the final sample. The degree of variability between the input and the final sample can be controlled by the amount of noise and is a hyperparameter of the method. The authors demonstrate the effectiveness of the method in a number of applications.

**Audience:**

Yes

**Claims And Evidence:**

Yes

**Requested Changes:**

Requested Changes:

1) For the qualitative results in Fig. 4, it looks like the visual quality of the samples becomes worse as the noise added in the input image increases (for instance at 70% some blurry artifacts are evident). It would be nice if the authors can include some quantitative results (in terms of FID) comparing the quality of the original images with the generated samples at different noising scales (or t_boom values).

2) For the results in Section 5, I think it might be worth including a experiment demonstrating the impact of augmenting with different t_boom on the classification accuracy for a small dataset like CIFAR-10/100.

3) For the super-resolution results, is there a general guideline for selecting t_boom for different downsampling factors? Intuitively, I would expect t_boom to be larger for larger downsampling factors since high-frequency details are added towards the end of the reverse process.

4) It is common to use deterministic samplers like DDIM, DPM-Solver etc. for fast diffusion model sampling. It might be worth adding an experiment demonstrating the robustness of the proposed method within existing experimental setup (for ex: super-resolution) in the context of these samplers.


**Strengths And Weaknesses:**

Strengths:

1) The presented method is simple and intuitive. Moreover, having a single hyperparameter and compatibility with pretrained-diffusion models makes adoption easy for practitioners.

2) The authors do a nice job of empirically validating the application of the method across a number of important application use-cases. More specifically, the data augmentation results in Section 5 demonstrate that augmenting datasets with purely synthetically generated samples might be suboptimal which is interesting.

Weaknesses: See Requested Changes

---

> ### Author Response · Authors · 2023-08-01
> **Addressing the changes**
>
> Thank you for reviewing our paper and the helpful feedback! Here is our response to your requested changes; we hope that you find them adequate.
>
> 1. You are right that the visual quality of the samples becomes worse. This is in part because real images are better than synthetic, even if one uses an excellent diffusion model such as Stable Diffusion. We have added a plot showing that FID increases as we increase $t_\text{BOOM}$ which we will add to the appendix. The values of FID are 0.0, 0.7, 0.97, 2.44, and 4.26 for $\frac{t_\text{BOOM}}{T}$ of 0%, 20%, 50%, 70%, and 80%.
>
> 2. We have added text and a figure to the appendix demonstrating the impact of augmenting with different $t_\text{BOOM}$ on the classification accuracy for a small dataset like CIFAR-100. The text reads: "The observed benefits of Boomerang in data augmentation presented in Section 5 are for specific values of $\frac{t_\text{BOOM}}{T}$. One can observe that there would be no benefit in picking  $\frac{t_\text{BOOM}}{T} = 0$\% because then no modification of the training data is performed and hence the augmented data will be the same as the non-augmented data, i.e., the training set. On the other hand, if we use $\frac{t_\text{BOOM}}{T} = 100$\%, we are essentially just sampling purely synthetic images from the diffusion model and using that for data augmentation. As shown in Section 5, there exists a value of $\frac{t_\text{BOOM}}{T}$ which is better than both of these extremes. In Figure 14 we show accuracy on the CIFAR-100 classification task for intermediate values of $\frac{t_\text{BOOM}}{T}$, demonstrating a tradeoff between accuracy and $\frac{t_\text{BOOM}}{T}$."
>
> 3. This is a good point, and in the latest version we added details in section 6.2 about general guidelines selecting $t_\text{BOOM}$ as well as the specific values of $t_\text{BOOM}$ that were effective in our superresolution/perceptual enhancement experiments. One specific change is that we wrote "In our examples, which used $1024 \times 1024$ images, we found $t_{\text{BOOM}}=50$ worked well with images downsampled by 4x while $t_{\text{BOOM}}=100$ worked best with images downsampled by 8x.  Our intuition suggests that a larger value of $t_{\text{BOOM}}$ is more appropriate for more degraded images, and our empirical results are in agreement."
>
> 4. We use different samplers throughout the paper which provides a reasonable amount of evidence that Boomerang works for different samplers. For Stable Diffusion we use PMDM, for CIFAR-10 we use FastDPM, and for CIFAR-100 we use a predictor-corrector sampler. Although it would be interesting to have an standalone experiment showing how different samplers affect Boomerang, we don't think it would fit the paper well, in part because we don't make any claims about the robustness of Boomerang to samplers. Please, let us know if you think this experiment is vital to the claims we make and why.

---

### Review · Reviewer_3PNA · 2023-08-06

**Summary Of Contributions:**

The authors propose a framework, termed Boomerang, that can be used to generate variations of a given image using a pre-trained diffusion model. The algorithm works by adding some amount of noise to the input image and then denoising the image using the reverse SDE initialized at the intermediate noise level. One can control how close the generated images will be to the original image by changing the amount of noise added. The authors show how this framework can be used to perform data augmentation, anonymization and image enhancement.

**Audience:**

Yes

**Broader Impact Concerns:**

I do not have any ethical concerns that are specific to this paper.

**Claims And Evidence:**

Yes

**Requested Changes:**

My main concerns were listed in the Section above. The main thing I would like to see is a discussion and a potential comparison with alternative methods for image variations that are widely used by the community, including SDEEdit and the image variations using CLIP guidance.

**Strengths And Weaknesses:**

I think the idea is very simple, yet effective. The applications of the framework are really important. Data anonymization becomes more and more important, especially as it becomes clear that diffusion models that are trained on LAION (such as Stable Diffusion) can memorize training images. The synthetic data application is also very important to avoid overfitting the dataset (in the case of small datasets) and to potentially mitigate biases in the model (e.g. by balancing different classes in the dataset). Finally, I really like the idea of using a pre-trained diffusion model to perform image enhancement without further training.

My main concern with this work is the novelty of the proposed method. I believe that the trick of producing image variations by adding noise and then denoising has been known to the community of people working with Stable Diffusion and other foundation models. For example, this feature seems to be supported out-of-the-box in this well-known open-source repo: [https://github.com/AUTOMATIC1111/stable-diffusion-webui/discussions/2918](https://github.com/AUTOMATIC1111/stable-diffusion-webui/discussions/2918).

Additionally, there are other ways to perform image variations too, e.g. by getting the CLIP embedding of the input image and then using it to guide the forward process. Adjusting the guidance scale has a similar effect to controlling the amount of noise added. I don't see any comparison with this method. Again, producing image variations seems to be something that the practitioners can already do and there are tools to do so, e.g. see [here](https://huggingface.co/spaces/lambdalabs/stable-diffusion-image-variations).

Finally, an alternative way to produce image variations is to use Dreambooth or some other framework for personalized text-to-image generation. I understand that this framework requires some additional work (finetuning the model) but the cost of this seems to be trivial with LoRA and other similar tricks widely used by the practitioners.

My overall comment is that I get the sense that producing image variations has been a topic of interest, at the very least to the community of practitioners, and I believe that the authors do not establish well enough the connection to the existing efforts. I tried to follow the references in these open-source implementations and it seems that many of them credit the SDEedit paper, which is very similar to the proposed framework. I appreciated the discussion of the differences with the SDEedit in the main paper, but I am not convinced that the idea is significantly different. The authors mention that in the SDEedit paper the authors run the whole reverse process with a modified noise schedule. However, the initialization of both methods is the same (a noisy version of the reference image) and even Boomerang can be though as SDEEdit with a modified noise schedule that spends no time in the first part of the reverse SDE. At the very least, I think that a comparison with SDEedit is required.

Another weakness of the paper is that for each one of the applications there are alternative ways to achieve them and there is no thorough comparison to these approaches. For example, there have been many recent works about diffusion models and synthetic data augmentation. It would be useful to thoroughly explain the differences to such methods and provide numerical comparisons. Further, data anonymization can be achieved by projecting an image to the latent space of a GAN (image inversion) and then perturbing this latent. Finally, the data anonymization might not work if the image to be anonymized was in the training set of the foundation model.

---

> ### Author Response · Authors · 2023-08-23
> **Comparing to other methods**
>
> Thank you for your review and feeback. It helped us run more experiments with Boomerang which are quite insightful and significantly improved our paper.
>
> **UnClip:** We compared to UnClip which is a Stable Diffusion model fine-tuned to accept CLIP guidance to produce variations of an input image. UnCLIP has a noise level parameter that behaves similarly to $t_\text{BOOM}$. We tried noise values of 0, 200, 400, 600, and 800. Unfortunately, we cannot add images to openreview posts, but the results were interesting and we will do our best to describe them here. We have made a figure (Figure 16 in the appendix) and added it to our current version of the paper demonstrating all of this. What happened is that the perceptual difference between the input image and any output image (even when noise=0) is quite large. *In other words, UnClip cannot recover the intial image.* This means that Boomerang seems to be better at smoothly interpolating between the initial image and the local samples, specifically when we want to sample really close to the input image.
>
> Moreover, Boomerang is faster than UnClip because we only have to do partial diffusion processes, speeding up the local sampling by 20-80% in our experiments.
>
>
> **Image Variations:** We also compared to mage Variations, which is also a Stable Diffusion model fine-tuned to accept CLIP guidance. This model has a guidance parameter instead of a noise paramter. We used guidance values of 15, 10, 5, 3, and 1. These images, like the UnClip results, do not sample very close to the initial image. It is a viable method for sampling more diverse points, but not if we want to sample very locally on the image manifold.
>
> Similarly to UnClip, Boomerang is faster than Image Variations, speeding up the local sampling by 20-80% in our experiments.
>
>
>
> **Dreambooth:** Although Dreambooth tries to make synthetic images which semantically resemble the subjects of the original images, it is quite different than Boomerang. First of all, it does not have a mapping which goes from image manifold to image manifold, which is the use case for Boomerang. Second, since Dreambooth by itself only learns a noise-to-image mapping, this means that Dreambooth cannot do perceptual resolution enhancement: after all, performing perceptual resolution enhancement requires being able to condition synthetic images on an existing (corrupted or downsampled) image. In order to perform local sampling, Dreambooth would necessarily require combination with an existing local sampling technique, such as Boomerang. Third, we cannot perform anonymization because Dreambooth is trying to make the semantics the same, i.e., perserve distinguishing characteristics. Fourth, Dreambooth requires finetuning a model which takes significantly more time than Boomerang, making it computationally infeasible for data augmentation. Therefore, we don't compare to Dreambooth.
>
>
> **SDEdit:** We acknowledge your observation concerning the resemblances in the underlying algorithmic procedures of both Boomerang and SDEdit. We have taken measures to set apart our approach from it.
>
> In the main paper, we have modified the initial paragraph of Section 7 to further highlight the fundamental distinctions between Boomerang and SDEdit. In essence, although both methods make use of the reverse diffusion process to achieve their goal, Boomerang differs in that it focuses on local sampling, unlike SDEdit that deals with image editing. Specifically, our objective is to generate variations of a natural image that are nearby on a learned manifold. This is in contrast to SDEdit, which relies on input sketches to create image variations. As a result, a meaningful comparison with SDEdit in the context of data anonymization, data augmentation, and image quality enhancement, requires obtaining sketches for the input images that would enable us to locally samples the image manifold. However, there is no evident automated method for obtaining such sketches. This limitation of SDEdit in performing local sampling further highlights the differences between Boomerang and SDEdit. We hope this clarifies the unique contributions of our work and provides a more comprehensive perspective.
>
> We have added a paragraph to in Section 7 to clarify this distrinction.

---

### Review · Reviewer_o4y3 · 2023-08-11

**Summary Of Contributions:**

In this paper, authors propose a simple technique to do "local sampling" of images from a pre-trained diffusion model where the term "local sampling" roughly translates to sampling images related to a given image. The idea is very simple: instead of inverting an image all the way to noise, you stop mid-way and return to the image manifold (giving the approach its name--Boomerang.) Since we return mid-way, the image does not completely lose its perceptual quality; however, since the generative process is stochastic, the generated image differs from the original image. Importantly, the simple idea can be implemented with minimal changes. The authors employ this simple technique for constructing privacy-preserving datasets, augmenting datasets, and enhancing image resolution.

**Audience:**

Yes

**Broader Impact Concerns:**

The most important concern is in terms of privacy-preserving applications. As far as I can see, there is no discussion on ensuring that the proposed method can preserve the user's privacy. Consider this scenario: a person deploys a system based on this technique where they take photos of users and "anonymize" them. However, they did not boomerang for long enough, and now the identity of the person in the photo is still revealed.

My concern may seem unwarranted to the authors. However, this technique is so simple and so easy to employ that anybody with access to a GPU can run this "anonymization" scheme. So, I believe there has to be a rigorous discussion about the claims of anonymity.

**Claims And Evidence:**

No

**Requested Changes:**

See the weaknesses section.

**Strengths And Weaknesses:**



### Strengths
- The technique is extremely simple and can be easily implemented by any practitioner to augment datasets and create similar-looking pictures without much overhead.
- It is important to demonstrate the applications in a paper with a simple technique. The authors nailed two important applications: preserving privacy and data augmentation; both are extremely important use cases.
- The empirical evidence for how this simple technique can generate samples that are close to the original image is competing.

### Weaknesses

- The biggest flaw I see is being unable to control any part of the generation. In the sense that you can not control what part of the image is being anonymized. Similarly, you can not control how (or if) each cascading boomerang enhances the image resolution; it will produce a similar image to the output of the previous cascade; however, you can not guide the generation towards enhanced resolution. Therefore, anonymization and super-resolution applications feel more like a stretch to me.

- Can the authors precisely clarify what are the changes between the anonymization and data augmentation experiments? In particular, if you took the data you used from augmentation experiments and trained only on augmented data, is the performance the same as the anonymized data?

- How do you decide if the image is being anonymized or prepared for data augmentation? Is the augmented data always anonymized? Is the anonymized data always augmented?

- Also, there are no error bars in any experiment. he absence of error bars is generally a concern. Could the authors comment on the lack of variance estimates? Isn't the data generation process stochastic and should lead to multiple answers no matter where you apply Boomerang?

- While data augmentation is the primary application, it is fairly limited by being unable to control what the diffusion model would generate. If someone wants to use Boomerang to create data, then adjusting the $t_{\mathrm{Boom}}$ parameter has a huge cost. You first sample a huge dataset with $t_{\mathrm{Boom}} = 100$. You train and check the performance. It did not work great. So, you do that again with $t_{\mathrm{Boom}} = 200$, and so on.

- Also, what is the synthetic data in Table 2? Is this the synthetic data generated from some pre-trained diffusion model? How do we know this generated synthetic data is relevant to the classification problem? This is more of curiosity than criticism. I want to understand the comparison.

- In Table 3, you mention that the number of cascades was selected based on the "best images." What are the best images? Who decides the best images?

- In Section 6.2, the last paragraph, you talk about the time it takes to generate an image with Boomerang. On what computing architecture are you reporting these numbers?

- In the perceptual resolution enhancement, can the authors comment on what fraction of the images enhanced using boomerang falls below the minimum distance threshold from Figure 5? Can this be a good metric to demonstrate super-resolution?

- The authors do a good job of not complicating a simple procedure. Well, for the most part. The purpose behind equation 7 was unclear. In the text, you state that the probability of $p(x_0')$ is the same as $p(x_0)$. Why is that true? Also, what is the inference from this small analysis? That the variability increases from increasing $t_{\mathrm{Boom}}$? Don't we already know this without the confusing math?



Overall, I feel that this paper is born out of a simple and beautiful observation. However, I wonder if this beautiful observation is as important as the authors claim. Clarifying the above questions can probably convince me otherwise.

---

> ### Author Response · Authors · 2023-08-23
> **Clarifying our work 1/2**
>
> Thank you for your review and feeback. It was very helpful in making our paper more clear. Since you asked for us to clarify the points above, we will do this in the hopes that it all makes sense! Please let us know if you have any additional questions.
>
> **Data augmentation and anonymization:**  In both cases, Boomerang is used on real data. In anonymization this is done for the end goal of anonymizing images so that the viewer doesn't know who, or what, was in the original image. We train a classifier on only this Boomerang data to display quantitative results on how anonymization affects classifcation performance. In data augmentation, we train a classifier and use Boomerang data along with the training dataset. To answer your question, if we took the data used for augmentation from the data augmentation experiments (the Boomerang data) and trained solely on that, yes this is the same thing as training on the anonymized data.
>
> In both data augmentation and anonymization we use Boomerang on the data. You could say that the data in data augmentation is anonymized, yes. The anonymization experiments in Table 1, however, do not have any data augmentation.
>
> **Error bars:** The error bars are not present because the standard error of the experiments in Figure 5 are just too small. To give you an idea, the largest error bars are 0.0017 for embedding distance and 0.21% for the number of images over the threshold. We have added a comment on this in the figure "The largest standard error is 0.0017 for embedding distance and 0.21\% for the number of images over the threshold."
>
> **Cost of Boomerang:** Yes you are right that there is a large cost in generating the data with Boomerang. This is because sampling from diffusion models is quite slow. However, with boomerang we don't have to sample all the way. Since we do partial sampling, we do have a significant speedup, 20-80% in our experiments, compared to using the full sampling routine.
>
> **Table 2:** The first row of Figure 2 is just normal CIFAR10 training data. The second row uses the pretrained FastDPM diffusion model to perform Boomerang on the images in the first row. We know that this model is relevant to the classification problem because this diffusion model is conditionally trained on CIFAR10.
>
>
> **Table 3:** We noticed a typo in that sentence so thank you for pointing this out. We have re-written the caption so that it is more clear and easy to understand. This is what we wrote: "For Cascaded results, the number of cascades were selected based on the best images as decided by a human observer who chose the image that had the best perceptual quality." The human observer in this case are the authors.
>
>
> **Computing hardware:** For our experiments described in Section 6.2 we use a Titan X GPU. We have clarified our text with the following sentence: "These times are reported for a single Nvidia GeForce GTX Titan X GPU."
>
>
> **Are we anonymizing via perceptual resolution enhancement?:** You asked this question and we ran an experiment seeing how many of the images are anonymized when we do perceptual resolution enhancment. We would like none of them to be anonymized obviously because we are just trying to restore the image to some higher perceptual quality, not change the identity of the subject. However, the lower resolution the image is, the more we are forced to anonymized because we have to Boomerang further. Still, with 8x upsampling, we end up anonymizing at most 32% of the images. This anonymization rate for perceptual enhancement is with respect to the unseen ground truth images, not the 8x downsampled image. This is true for cascade values of 1 through 10 as can be seen in the table:
>
>
> |N_\text{casc}|% Anonymized|standard error|
> |-|-|-|
> |1|29.5% |0.658%|
> |2|29.9%|0.663%|
> |3|31.3%|0.680%|
> |4|30.7%|0.673%|
> |5|30.8%|0.674%|
> |6|31.7%|0.685%|
> |7|29.2%|0.654%|
> |8|30.6%|0.672%|
> |9|31.1%|0.678%|
> |10|29.2%|0.654%|
>
> This is quite surprising because we are not doing anything to mitigate anonymization yet most of the images not not anonymized An interesting future direction for this work is to do perceptual resolution enhancement without anonymizing. Thank you for the suggestion!

---

> > ### Author Response · Authors · 2023-08-23
> > **Clarifying our work 2/2**
> >
> > **Math:** We appreciate that you brought to our attention the need for further clarification regarding Equation (7). As you correctly noted, Equation (7) offers a mathematical perspective on the increase of variability in images as $t$ increases; due to markdown limitations in openreview we use $t$ instead of $t_\text{BOOM}$. While this is intuitive, we believe that this equation provides a more formal justification.
> >
> > We have included the following discussion in the main paper.
> >
> > "Recall that $p(\mathbf{x}_0') $ is, by definition, the distribution of images acquired through the Boomerang method. However, unlike $p(\mathbf{x}_0' | \mathbf{x}_t)$,  it is not conditioned on a specific image at diffusion step $t$. Specifically, $p(\mathbf{x}_0')$ is the distribution of images generated by the diffusion model when the reverse process is initiated at step $t$ using noisy images obtained from $\mathbf{x}_t' = \sqrt{\alpha_t} \mathbf{x}_0 + \sqrt{1 - \alpha_t} \boldsymbol{\epsilon}$, where the original images $\mathbf{x}_0$ are drawn from $p(\mathbf{x}_0)$ and $\boldsymbol{\epsilon} \sim \mathcal{N}(\mathbf{0}, \mathbf{I})$. The distribution of these noisy images is equivalent to $\mathcal{N}(\sqrt{\alpha_t} \mathbf{x}_0, 1 - \alpha_t \mathbf{I})$, with $\mathbf{x}_0 \sim p(\mathbf{x}_0)$. This distribution is equal to the forward diffusion process distribution at step $t$, denoted as $q(\mathbf{x}_t | \mathbf{x}_0)$ (recall Equation (2) in the main text). Given that the diffusion model is well-trained, we can expect that its output matches the original image distribution regardless of which step the reverse process in initiated, as long as the same forward diffusion process noise schedule is used."
> >
> >
> > We hope that we have answered all your questions well. Let us know if we should elaborate more. Thank you!

---

### Decision · Action_Editor_kt8r · 2023-10-25

**Recommendation:** Accept with minor revision

**Comment:**

Two of the three reviewers acknowledged that the author response and paper revision has addressed most of the concerns raised in the original reviews and are leaning towards acceptance.

One outstanding concern from the third reviewer is about the applicability of the method for anonymization applications and lack of sufficient experimental support for this - which I agree with. In particular it is not clear how the proposed method can anonymize sensitive attribute while retaining other attributes of interest since the only controllability it allows is through controlling the variance of the noise added to the data. I agree with this concern and suggest the author revise the paper to clearly describe the limitations of the method when used for the anonymization task.

**Audience:**

As pointed out by all three reviewers, the method proposed in the paper is simple to implement on top of pretrained diffusion models which makes it an attractive candidate for practitioners.

**Claims And Evidence:**

The paper proposes Boomerang, a method to do local sampling on the image manifold. Given an image, the method runs forward diffusion up to certain steps to get a noisy image and then runs the reverse process on this noisy instance to get a new image near the original data point. The main claims of the paper are centered around proposing a method that is able to locally sample the image manifold (that is implicitly parameterized by a pretrained diffusion model) near the given data point. The paper shows the applicability of the proposed method in three settings: generating privacy preserving / anonymized datasets, perceptual image enhancement of low resolution images, and data augmentation for supervised classification tasks.

All three reviewers appreciate the simplicity of the method. There was a concern from reviewer 3PNA about novelty of the method over existing approaches already implemented in open-source repos and lack of comparisons with other relevant methods for image-variations, which the authors have addressed in the author-response and paper revision by additional experiments.

One outstanding concern from reviewer o4y3 is about the lack of convincing experiments to justify the use of Boomerang in the anonymization and image enhancement applications. I tend to agree with this observation from the reviewer, particularly for the anonymization task. Since the authors don't evaluate the method on if/how it meets specific anonymization criteria established in the privacy-preserving literature about removing sensitive attributes, it would make sense to dilute the claims on the anonymization application and also add some context on the existing work on privacy preserving methods.

---

> ### Author Response · Authors · 2023-11-16
> **Thank you**
>
> Thank you for the feedback and the recommendation. We have modified the paper to address your concern. That is, we clarified that Boomerang is able to coarsely change level of anonyminity but is not able to anonymize specific attributes. These are the parts that we changed:
> 1. "Boomerang local sampling enables privacy-preserving machine learning by generating data points that are similar to real (i.e., sensitive) data, while being dissimilar enough to existing data so as to not leak any ground-truth sensitive information, in the case of a membership inference attack for example."
> 2. "Boomerang can anonymize entire datasets to varying degrees controlled by the hyperparameter $t_\text{BOOM}$."
> 3. "Additionally, we anonymize natural images; specifically, we define that a natural image $\mathbf x_0$ is anonymized to $\mathbf x_0'$ if the  features of each image are visibly different such that an observer would guess that the two images are of different objects."
> 4. "Therefore, we qualitatively and quantitatively establish that Boomerang anonymization is a controllable, efficient manner of anonymizing images by removing identifiable features."
>
> Which we changed to:
>
> 1. "Boomerang local sampling enables privacy-preserving machine learning by generating data points that are similar to real (i.e., sensitive) data, **yet not the same. The degree of similarity to the real data can be coarsely controlled through the $t_\text{BOOM}$ parameter, however Boomerang can not remove specific sensitive attributes while retaining other attributes.**"
> 2. "Boomerang can anonymize entire datasets to varying degrees controlled by the hyperparameter $t_\text{BOOM}$, **which coarsely defines the anonymization level.**"
> 3. "Additionally, we anonymize natural images. Specifically, we define that a natural image $\mathbf x_0$ is anonymized to $\mathbf x_0'$ if the  features of each image are visibly different such that an observer would guess that the two images are of different objects **(note that we do not control which features are being anonymized).**"
> 4. "Therefore, we qualitatively and quantitatively establish that Boomerang anonymization is **an efficient method of anonymizing images by local sampling.**"
>
> The updated camera ready paper will be uploaded shortly. Thank you again.